# The Hippocampal Place Field Gradient:
# A Bio-inspired Framework Building Multiscale Representation for Better Sample Efficiency

**Shujun Zhou** [* 1 2]  **Junrong Qi** [* 3]  **Guozhang Chen** [3]

## Abstract

The hippocampus encodes space through a striking gradient of place field sizes along its dorsal-ventral axis, yet the principles generating this continuous gradient from discrete grid cell inputs remain unclear. We propose a unified theoretical framework establishing how multiscale hippocampal place fields arise from the frequency-dependent decay of grid cell projections. Functionally, this organization establishes an inductive bias in the population code, managing a fundamental trade-off between spatial precision and sample efficiency. Translating this insight to artificial neural networks, we incorporate a hippocampus-inspired positional embedding (HIPE) into the Transformer architecture to induce multi-scale representation. Experimental results confirm that this mechanism effectively improves data efficiency. Our work establishes a link between neural connectivity, activity patterns, and learning, suggesting a principled approach to utilizing multi-scale representations for sample-efficiency learning. Our codes are available at `https://github.com/AIogry/relative_PE`.

## 1. Introduction

The hippocampus and entorhinal cortex are brain structures renowned for their roles in spatial navigation and data

Work primarily completed while Shujun Zhou was an undergraduate student at Peking University, supervised by Guozhang Chen. [*]Equal contribution [1]Yuanpei College, Peking University, Beijing, China [2]Department of Neuroscience, The University of Texas at Austin, Austin, USA [3]State Key Laboratory of Multimedia Information Processing and the National Engineering Research Center of Visual Technology, School of Computer Science, Peking University, Beijing, China. Correspondence to: Guozhang Chen <guozhang.chen@pku.edu.cn>.

*Proceedings of the 43rd International Conference on Machine Learning*, Seoul, South Korea. PMLR 306, 2026. Copyright 2026 by the author(s).

efficient learning (Liao & Losonczy, 2024). Place cells in hippocampal CA1 mark specific locations like neural landmarks, while grid cells in the medial entorhinal cortex (MEC) create a periodic, spatially tiling pattern (Hafting et al., 2005). They cooperate to construct neural representations of both spatial and potentially non-spatial cognitive domains (Behrens et al., 2018; Caitlin S. Mallory et al., 2018).

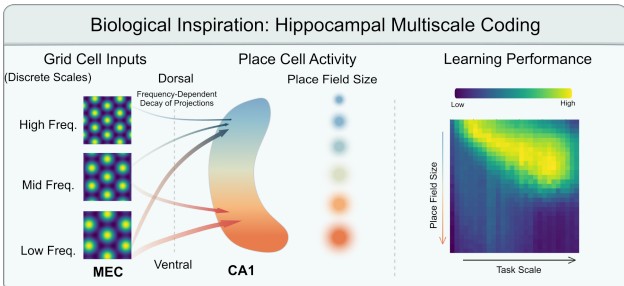

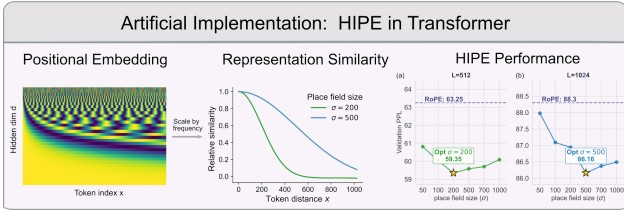

*Figure 1.* An overview of our framework in linking structural connectivity, multiscale coding and learning in biological and artificial systems.

Despite this established relationship, a key puzzle, as highlighted by Ref. (Strange et al., 2014), persists: how does the hippocampus transform inputs from discretely scaled grid cell modules (Stensola et al., 2012) into the observed *smooth, continuous* gradient of place field sizes along its dorsal-ventral axis (Kjelstrup et al., 2008; Keinath et al., 2014)? Moreover, what functional advantages does this specific multiscale organization confer? While prior theoretical works have established grid cells' mathematical role in the construction of place fields (Solstad et al., 2006; Sorscher et al., 2023), critical gaps remain in: (1) explaining how continuous place field gradients emerge from discrete modules (2) linking this gradient to learning advantages, and (3) applying this to inform the design of deep learning

architectures.

In this paper, we propose that the hippocampal place field gradient is a sophisticated architectural design that optimizes learning across varied conditions by shaping the inductive biases of the neural code. We present a unified theoretical framework that bridges anatomical projections, coding geometry, and learning to resolve these longstanding questions (Figure 1). Specifically, we make three primary contributions. First, we analytically derive how the frequency-dependent decay of projection weights determines place field size, explaining how the continuous gradient of place field size arises from discrete grid modules as observed in the hippocampus. Second, we use population coding theory to show how place field size shapes the trade-off between precision and generalization, identifying the optimal place field width for few-shot learning and how this optimality depends on the structure of the task environment. Third, bridging these biological insights with modern deep learning, we introduce Hippocampus-Inspired Positional Embedding (HIPE). By translating the frequency-dependent grid-to-place projection into a spectral scaling transformation on rotary positional embeddings, we demonstrate that this bio-inspired inductive bias significantly improves the learning efficiency of Transformer models.

## 2. Mathematical Model for the Emergence of Place Field Gradient

To understand the relationship between place codes and grid codes, we construct a mathematical model to analytically derive how a specific grid-place cell projection pattern best generates place fields with desired sizes.

### 2.1. Grid-to-Place Cell Mapping Determines Place Field Size

We denote the tuning function of place cell $i$ as $p_i(x)$, where here $x$ is the location. To simplify the mathematical formulation, we consider $x$ in 1D space, but the results can be extended to $d$-dimensional space. The activity of $n_p$ place cells and $n_g$ grid cells can be represented as $\boldsymbol{p}(x) = [p_1(x), p_2(x), \cdots, p_{n_p}(x)]^\top$ and $\boldsymbol{g}(x) = [g_1(x), g_2(x), \cdots, g_{n_g}(x)]^\top$ respectively. We now formalize how the characteristic scale of hippocampal place fields emerges from the linear transformation mapping grid cell inputs to place cell outputs within a spatial domain of length $L$. The periodic grid-like firing patterns observed in medial entorhinal cortex grid cells (Hafting et al., 2005) can be modeled as trigonometric functions:

$$g_{2k-1}(x) = \cos\left(\frac{2\pi k}{L}x\right), \quad g_{2k}(x) = \sin\left(\frac{2\pi k}{L}x\right),$$

where $k = 1, 2, \cdots, n_g/2$. Here, $g_{2k-1}(x)$ and $g_{2k}(x)$ share the same spatial frequency, which can be regarded as

the grid module with spatial frequency $k$. As for the tuning functions of place cells, we use a Gaussian profile as in (Grijseels et al., 2021; Rooke et al., 2024):

$$p_i(x) = p(x; \mu_i) = \frac{A}{\sqrt{2\pi}\sigma_p} \exp\left[-\frac{(x - \mu_i)^2}{2\sigma_p^2}\right], \quad (1)$$

where $\mu_i \sim \mathcal{U}(0, L)$ specifies the field center and $\sigma_p$ denotes the place field width. Consider the linear mapping from grid cells to place cells connected by the weight matrix $\boldsymbol{W}$ with size $n_g \times n_p$ as in (Sorscher et al., 2023; Solstad et al., 2006), the full population response is expressed as:

$$\boldsymbol{p}(x) = \boldsymbol{W}_{n_p \times n_g} \, \boldsymbol{g}(x). \quad (2)$$

Given the tuning curves of the place cells, the optimal weight matrix from grid-place cell mapping is the one that minimizes the error between $\boldsymbol{p}$ and $\boldsymbol{Wg}$. This corresponds to solving the least-squares minimization of the reconstruction error:

$$\min_{\boldsymbol{w}_i} \int_0^L dx \, |p_i(x) - \boldsymbol{w}_i \cdot \boldsymbol{g}(x)|^2.$$

where $\boldsymbol{w}_i$ represents the $i^{\text{th}}$ row of the weight matrix $\boldsymbol{W}$. The orthogonality of the grid cell tuning functions $(g_j(x))$ over the interval $[0, L]$ decouples the problem into minimizing each element of $\boldsymbol{w}_i$ independently. The reconstruction error is minimized when $W_{ij}$ equals to the projection of the target place cell's tuning curve, $p_i(x)$, onto the corresponding grid cell's tuning curve, $g_j(x)$:

$$W_{ij} = \frac{2}{L} \int_0^L dx \, p_i(x) g_j(x) \quad (3)$$

For Gaussian place fields, as the place field width $\sigma_p$ is smaller than the scale of the environment, the projection can be approximated by a closed-form expression (Figure 2a):

$$W_{i,2k-1} = \frac{2}{L} \int_{-\infty}^{+\infty} dx \, \cos\left(\frac{2\pi k}{L}x\right) \frac{A}{\sqrt{2\pi}\sigma_p} e^{-\frac{(x-\mu_i)^2}{2\sigma_p^2}}$$
$$= \frac{2A}{L} e^{-\frac{\sigma_p^2(2\pi k/L)^2}{2}} \cos\left(\frac{2\pi k}{L}\mu_i\right). \quad (4)$$

$$W_{i,2k} = \frac{2}{L} \int_{-\infty}^{+\infty} dx \, \sin\left(\frac{2\pi k}{L}x\right) \frac{A}{\sqrt{2\pi}\sigma_p} e^{-\frac{(x-\mu_i)^2}{2\sigma_p^2}}$$
$$= \frac{2A}{L} e^{-\frac{\sigma_p^2(2\pi k/L)^2}{2}} \sin\left(\frac{2\pi k}{L}\mu_i\right). \quad (5)$$

Thus, to generate place cells with a desired place field size $\sigma_p$, the projection strength from the grid module with frequency $k$ to place cell $i$ ($|W_i(k)| = \sqrt{W_{i,2k-1}^2 + W_{i,2k}^2}$) should decay exponentially with spatial frequency according to the place field size $\sigma_p$ (Figure 2b):

$$|W_i(k)| \propto \exp\left[-\frac{\sigma_p^2(2\pi k/L)^2}{2}\right]. \quad (6)$$

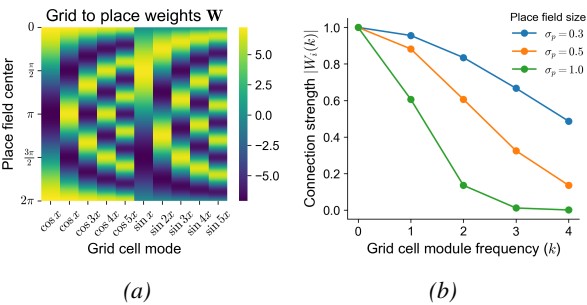

*(a)*          *(b)*

*Figure 2.* (a) Grid-to-place cell weights as a function of place field center $\mu$. (b) Projection strength from grid cells of varying spatial frequencies $k$ to place cells ($|W_i(k)|$), demonstrating the exponential attenuation with $k$. Environment length $L = 2\pi$.

This has a clear theoretical implication: despite the fact that grid codes themselves operate on discrete spatial scales, continuous tuning of place field size can be achieved by systematically adjusting the decay of weights across spatial frequencies. **Broader place fields** (larger $\sigma_p$) preferentially recruit low-frequency grid modules, producing coarse spatial representations, while place cells with **narrower place fields** requires input from both low and high-frequency modules to obtain a fine-grained spatial detail.

## 2.2. Predictions for Anatomical Projection Patterns

To account for the empirically observed dorsoventral gradient in hippocampal place field sizes (Kjelstrup et al., 2008), our model predicts a corresponding gradient in anatomical projections: first and foremost, compared to ventral place cells, dorsal CA1 and CA3 place cells should receive more inputs from the dorsal MEC grid cells. This is already supported by the distance-dependent wiring patterns of neurons (Gandolfi et al., 2023). Moreover, to generate desired place fields in all areas of CA1 and CA3, dorsal CA1 and CA3 place cells should integrate inputs from both dorsal (small-scale) and ventral (large-scale) MEC modules, whereas ventral CA1/CA3 should predominantly receive input from ventral MEC. This hypothesis invites direct empirical evaluation on the heterogeneity of MEC to CA1/CA3 projections along the dorsal to ventral axis using recently available high-resolution hippocampal projectome datasets (Qiu et al., 2024).

# 3. Multiscale Place Field Underlies Multiscale Learning

In light of this anatomical gradient in place field sizes, a natural question arises: What functional significance does this graded variation hold for cognitive processes? Experimental studies have shown that while place field sizes vary systematically along the hippocampal dorsal-ventral axis (Kjelstrup et al., 2008), the population code remains highly redundant, with a small ensemble (e.g., 20 - 100) of CA1

neurons achieving spatial localization with remarkable precision (Wilson & McNaughton, 1993; Hazon et al., 2022). Despite this redundancy, the functional implications for such a graded variation in place field size remains unclear. We explore this question through the lens of code-task alignment, investigating how the structural properties of the place code influence learning performance across environments.

## 3.1. The Functional Roles of Large- and Small-Scale Place Fields

To illustrate the functional implications of place field size in learning, we designed a context-dependent computation task (Figure 3) using the grid-to-place linear mapping framework (Section 2.1). In this task, the agent is required to perform either an "AND" or "OR" operation depending on its spatial region. The model is trained on a limited set of spatial locations and evaluated on a uniformly sampled test set. We considered two different training scenarios to probe the system's performance: a few-shot learning regime, where the model is trained with a small number of samples to assess generalization, and a data-rich regime, where the model is trained with many samples to evaluate its final precision.

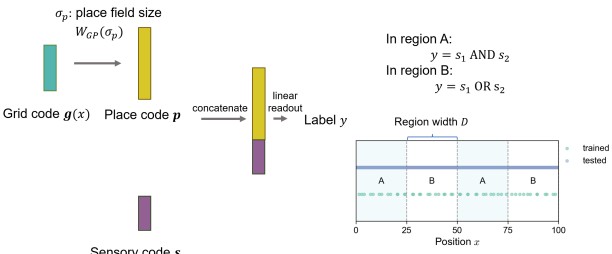

*Figure 3.* Model architecture and context-dependent task design.

As the number of training samples increases, test accuracy improves across all place field widths $\sigma_p$ (Figure 4). However, the relative advantage between different place field sizes shifts: **when sample size is large, narrow (small-scale) place fields yield better precision; when samples are few, broader (large-scale) place fields outperform by facilitating generalization.** This highlights a fundamental trade-off: while small fields guarantee high-fidelity learning under data-rich regimes, large fields enable few-shot learning, offering a more sample-efficient pathway for navigating expansive environments.

## 3.2. Optimal Place Field Size Depends on Task Structure

While our earlier results reveal a general trade-off between precision and generalization governed by place field size, a natural question arises: is there an *optimal* place field width that maximizes learning performance for a given task? To address this, we systematically varied the spatial scale of the

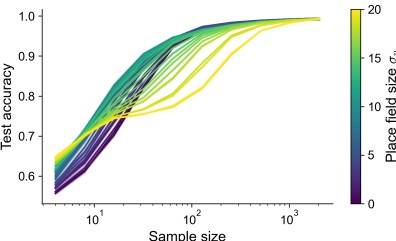

*Figure 4.* Test accuracy under varying training sample sizes and place field widths. The length of the maze $L = 100$ and region width $D = 25$. When sample size is large, narrow (small-scale) place fields yield better precision; when samples are few, broader (large-scale) place fields outperform by facilitating generalization.

task environment and evaluated how the best-performing place field size changes.

Remarkably, we find that under few-shot learning conditions, there exists a specific place field width that yields the highest test accuracy (Figure 5a, Appendix A for non-linear activation function). Further analysis shows that this optimal width scales proportionally with the characteristic region size $D$ of the environment (Figures 5b, 6). In other words, the best-performing hippocampal code is not fixed but adapts to the spatial demands of the task — larger environments or coarser tasks favor broader place fields, while fine-grained tasks benefit from narrower, high-resolution fields. This finding suggests that optimal learning requires

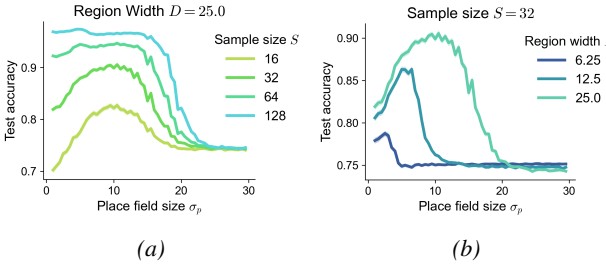

*(a)*          *(b)*

*Figure 5.* (a) Relationship between test accuracy and place field width. (b) Optimal place field width scales linearly with region width $D$. The results are the average of 1000 repeats with shaded areas showing the S.E.M..

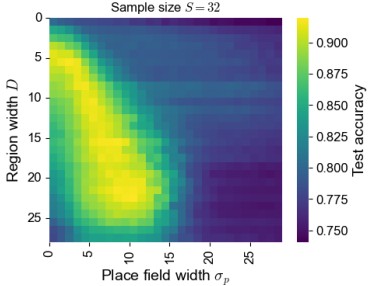

*Figure 6.* Performance landscape under varying place field sizes and region widths.

a dynamic alignment between neural code and task structure, where the inductive bias of the population code is

tuned to the problem's scale. Such adaptive tuning may explain why the hippocampus exhibits a gradient of place field sizes along its dorsal-ventral axis: by distributing place cells across a range of scales, the system ensures that at least some subpopulations are well-matched to the structure of the current task or environment. Importantly, this multiscale architecture could enable the hippocampus to support robust few-shot generalization across diverse spatial and non-spatial contexts.

### 3.3. Understanding Optimal Place Field Size Through Population Code Structure

To explain the observed trade-off between precision and generalization, and the existence of optimal place field widths, we adopt the theoretical framework developed by Bordelon et al. (Bordelon & Pehlevan, 2022) to examine how place field width $\sigma_p$ influences learning performance, as quantified by the generalization error:

$$E_g^{\text{full}} = \langle (\mathbf{w}_s \mathbf{s}(x) + \mathbf{w}_p \mathbf{p}(x) - y(x))^2 \rangle$$
$$= \langle (\mathbf{w}_p \mathbf{p}(x) - y(x))^2 \rangle + \langle \mathbf{w}_s \mathbf{s}(x)(2y(x) - \mathbf{w}_s \mathbf{s}(x)) \rangle.$$

where $y(x)$ is the target function and $\mathbf{w} \cdot \mathbf{p}(x) + \mathbf{w}_s \cdot \mathbf{s}(x)$ is the learned model output. Since the sensory code $\mathbf{s}(x)$ and the place code $\mathbf{p}(x)$ are orthogonal by construction in this task, the weights $\mathbf{w}_s$ and $\mathbf{w}_p$ can be optimized independently. The sensory weights $\mathbf{w}_s$ can perfectly fit the sensory component of the task, so the remaining generalization error is entirely determined by the place code term $\langle (\mathbf{w}_p \mathbf{p}(x) - y(x))^2 \rangle$. Thus we only analyze this remaining term, which reflects the contribution of place code to generalization error:

$$E_g = \langle (\mathbf{w} \cdot \mathbf{p}(x) - y(x))^2 \rangle$$

By decomposing $y(x)$ and $\mathbf{w} \cdot \mathbf{p}(x)$ onto grid codes $(y(x) = \sum_k \nu_k g_k(x), \mathbf{w} \cdot \mathbf{p}(x) = \sum_k \hat{\nu}_k g_k(x))$, the expected generalization error for a training set of sample size $S$ can be approximated as (see Appendix B):

$$E_g = \frac{\kappa^2}{1 - \gamma} \sum_k \frac{\nu_k^2}{(\lambda_k S + \kappa)^2}, \tag{7}$$

$$\kappa = \alpha + \kappa \sum_k \frac{\lambda_k}{\lambda_k S + \kappa}, \tag{8}$$

$$\gamma = S \sum_k \frac{\lambda_k^2}{(\lambda_k S + \kappa)^2}, \tag{9}$$

where $\alpha$ is the L2-regularization coefficient, and $\{\lambda_k\}$ are the eigenvalues of the spatial covariance function of place cell activity $\Sigma(x, x') = \mathbf{p}(x)^\top \mathbf{p}(x')$. For the model illustrated in Figure 3, the eigenvalues of the covariance function (see Appendix A) are

$$\lambda_k = \exp\left[ -\sigma_p^2 k^2 \left( \frac{2\pi}{L} \right)^2 \right]. \tag{10}$$

Equation 7 reveals that generalization error depends on three key factors: the sample size $S$; the alignment between the task and code, reflected in the coefficients $\{\nu_k\}$; and the spectral structure of the code, determined by $\{\lambda_k\}$. Specifically, the contribution of each mode to the error decreases with $\lambda_k$ and increases with $\nu_k$. Thus, under fixed total variance, minimizing generalization error requires concentrating variance on modes aligned with the task. This demonstrates the principle of **code-task alignment**: optimal learning performance emerges when the representational code and the task share a matched spectral structure.

To gain analytical insight into how the generalization error $E_g$ and the optimal place field size $\sigma_p$ depend on task scale , we approximate the task spectrum as being entirely concentrated at a single mode $n = \frac{L}{2D}$, i.e., $\nu_k = \nu_n \delta_{k,n}$ (where $\nu_n$ represents the magnitude of the projection onto this dominant mode, and $\delta_{k,n}$ is the Kronecker delta). Substituting this approximation into the expression for $E_g$ (Equation 7-9) and set $\alpha = 0$ (no L2 regularization) yields:

$$E_g = \frac{\kappa^2}{1-\gamma}\frac{\nu_n^2}{(e^{-\sigma_p^2 n^2 (2\pi/L)^2}S + \kappa)^2} \tag{11}$$

$$\approx \frac{\nu_n^2}{1-\gamma}\frac{1}{(C\sigma_p e^{-\sigma_p^2 n^2(2\pi/L)^2} + 1)^2} (S \ll \kappa), \tag{12}$$

where $C = \sqrt{2\pi}S/L$ . Therefore, we can find the optimal place field size under the few-shot learning regime by minimizing $E_g$, and this corresponds to maximizing $\sigma_p e^{-\sigma_p^2 n^2 (2\pi/L)^2}$ with respect to $\sigma_p$:

$$\sigma_p^{\text{opt}} = \arg\min_{\sigma_p} E_g = \frac{L}{2\sqrt{2\pi}n} \sim L \cdot n^{-1} \sim D. \tag{13}$$

which is independent of sample size $S$ (for a full derivation, see Appendix C). This explains the empirical findings where the optimal place field size $\sigma_p$ given a specific training data size $S$ is approximately the optimal place field size for all $S$ in the few-shot learning regime.

### 3.4. A Multi-Scale Place Code Ensures Robust Few-shot Learning Performance

The preceding analysis demonstrates that for any single task, an optimal place field size exists that maximizes learning efficiency by aligning the code's inductive bias with the task's spatial structure. However, a biological system must operate effectively across a multitude of environments with varying statistical properties. This raises a critical question: how can the brain achieve robust performance without knowing the scale of the task in advance?

We hypothesize that the observed heterogeneity of place field sizes along the hippocampal axis provides the functional advantage. To test this, we simulated a model endowed with a multi-scale place code, where place field widths were drawn from a uniform distribution ($\sigma_p \sim \mathcal{U}(5,20)$), and compared its performance against single-scale models across tasks with different region widths. The results, shown in Figure 7, reveal a distinct advantage for the multi-scale architecture. While a finely-tuned single-scale model can achieve peak performance on a specific task, the multi-scale model consistently performs well across all environments, achieving a higher average accuracy than any single-scale counterpart ($p < 10^{-4}$, independent t-test, Bonferroni corrected). This demonstrates that by maintaining a diverse repertoire of spatial scales, the system is prepared to learn efficiently regardless of the environment's specific structure. This provides a compelling functional rationale for the continuous gradient of place field sizes observed in the hippocampus (Kjelstrup et al., 2008), suggesting it as a deliberate design principle that enables flexible and rapid learning in novel or changing conditions. Exploring this link between representational diversity and adaptive learning is a crucial step toward understanding the versatility of biological intelligence.

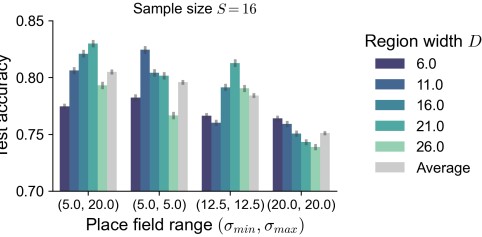

*Figure 7.* The functional advantage of a multi-scale place code architecture under few-shot learning regime (sample size $S = 16$). Each model is tested on five tasks with different region widths with 1000 repeats (error bar shows the S.E.M.) The multi-scale model, particularly $\sigma_p \sim \mathcal{U}(5,20)$, demonstrates consistently high performance and achieves a superior average accuracy across all environments compared to any single-scale model.

## 4. Translating Biological Inductive Bias to Transformers

We next translate the aforementioned biological inductive bias into Transformers by proposing hippocampus-inspired positional embedding (HIPE), a multiscale positional encoding framework built on the rotary frequency basis. Standard rotary positional embeddings (RoPE) (Su et al., 2024) serve as the unscaled sharp-field reference, while nonzero-$\sigma_p$ HIPE applies frequency-dependent spectral modulation to induce broader positional kernels.

Within the HIPE family, positional information is represented through periodic rotations. The phase-coding structure of these rotations functionally resembles the periodic coding mechanism of grid cell modules, providing a natural computational bridge between biological grid-like bases and Transformer positional encodings. HIPE then transfers

the grid-to-place principle derived above into attention by applying frequency-dependent spectral modulation.

## 4.1. RoPE as the Unscaled Sharp-Field

RoPE encodes position by dividing the embedding vector of dimension $d$ into $d/2$ consecutive pairs of dimensions. We define the RoPE operation $f(\mathbf{x}, m)$ as applying a rotation to the vector $\mathbf{x}$ at position $m$ in each of these 2D subspaces:

$$f(\mathbf{x}, m) = \mathbf{R}_{\Theta, m} \mathbf{x} \tag{14}$$

where $\mathbf{R}_{\Theta, m} \in \mathbb{R}^{d \times d}$ is a block-diagonal rotation matrix formed by the discrete frequencies $\{\omega_j\}_{j=1}^{d/2}$ (see Appendix D). In each subspace $j$, treating the vector pair as a complex number $x_j$, the rotation corresponds to multiplication by $e^{im\omega_j}$. The positional attention score between position $m$ and position $n$ for query vector $\mathbf{q}$ and key vector $\mathbf{k}$ is then

$$\langle f(\mathbf{q}, m), f(\mathbf{k}, n) \rangle = \sum_{j=1}^{d/2} \mathrm{Re} \left[ q_j k_j^* e^{i\omega_j(m-n)} \right] \tag{15}$$

Crucially, RoPE preserves the constant amplitude of all discrete rotary frequency components. In our hierarchy, this unscaled spectrum serves as the sharp-field reference: all available rotary frequencies contribute without additional attenuation, preserving fine-grained relative positional discrimination. This provides the dorsal-limit analogy used in our biological interpretation.

## 4.2. HIPE: Frequency-Dependent Spectral Scaling

We now derive the spectral scaling rule used by HIPE. To isolate the positional inductive bias introduced by the embedding mechanism, we consider the case where the query and key are semantically identical, i.e., $\mathbf{q} = \mathbf{k}$. Under this condition, Eq. 15 reduces to a summation of cosine waves weighted by the semantic energy in each frequency subspace:

$$\mathrm{Score}(\Delta) = \sum_{j=1}^{d/2} \|\mathbf{q}_j\|^2 \cos(\omega_j \Delta) \tag{16}$$

where $\|\mathbf{q}_j\|^2$ is the squared Euclidean norm of the $j$-th query subspace, and $\Delta = m - n$ denotes the relative distance between two positions.

To obtain a stable positional kernel, we separate the positional mechanism from input-dependent semantic variation. Specifically, we treat the semantic energy $\|\mathbf{q}_j\|^2$ as independent of the frequency index $j$. Under an isotropic energy assumption, $\mathbb{E}[\|\mathbf{q}_j\|^2]$ is approximately constant across subspaces. HIPE then applies a frequency-dependent scaling factor $S(\omega_j)$ to both query and key vectors. Since this scaling is applied symmetrically to both sides of the dot product,

the induced spectral weight becomes $S(\omega_j)^2$, yielding the positional kernel

$$\mathrm{Attn}_{\mathrm{pos}}(\Delta) \approx \sum_{j=1}^{d/2} \underbrace{S(\omega_j)^2}_{\text{Synaptic Weight}} \cdot \underbrace{\cos(\omega_j \Delta)}_{\text{Grid Code}} \tag{17}$$

This equation mirrors the grid-to-place integration model in Section 4.1: the cosine term plays the role of a periodic grid-like basis, while $S(\omega_j)^2$ determines how strongly each frequency contributes to the resulting place-like positional kernel. Our goal is to choose $S(\omega)$ such that the spectral summation approximates a Gaussian field profile,

$$\sum_{j=1}^{d/2} S(\omega_j)^2 \cos(\omega_j \Delta) \propto e^{-\frac{\Delta^2}{2\sigma_p^2}} \tag{18}$$

To satisfy Eq. 18, the spectral weights $S(\omega)^2$ should match the Fourier transform of the target Gaussian profile. However, RoPE samples frequencies exponentially, $\omega_j \propto \theta_{\mathrm{base}}^{-2j/d}$, rather than uniformly. This induces a non-uniform frequency density proportional to $1/\omega$. To make the discrete summation approximate the corresponding continuous Gaussian integral, we include a Jacobian compensation term. The resulting scaling factor applied to the query and key vectors is

$$S(\omega) = \lambda \cdot \sqrt{\omega \cdot \exp\left(-\frac{\sigma_p^2 \omega^2}{2}\right)} \tag{19}$$

where $\lambda$ is a normalization constant.

This expression has three interpretable components. First, the Gaussian decay term $\exp(-\sigma_p^2 \omega^2 / 2)$ follows the frequency-dependent attenuation derived from the grid-to-place projection in Eq. 6, with $\omega$ corresponding to the spatial frequency $2\pi k/L$. Larger $\sigma_p$ suppresses high-frequency components more strongly, producing broader positional fields. Second, the factor $\sqrt{\omega}$ compensates for the non-uniform density of RoPE frequencies. Third, the normalization constant $\lambda$ maintains the expected embedding energy and prevents numerical instability during training.

## 4.3. Layer-wise Deployment and Efficient Implementation

HIPE is a general framework rather than a single fixed positional encoding. In principle, different layers or frequency groups may use different field scales $\sigma_p$. In this work, we use a simple bipartite schedule as a controlled instantiation of this family: shallow layers use standard RoPE within the HIPE framework, while deeper layers use a larger nonzero $\sigma_p$ to promote broader positional integration, a design intended to improve generalization in sample-limited regimes.

This design reflects the functional hierarchy suggested by the hippocampal dorsal–ventral axis. Shallow layers preserve fine-grained positional discrimination, analogous to small dorsal place fields, whereas deeper layers aggregate information over broader ranges, analogous to larger ventral place fields. Importantly, the use of RoPE in shallow layers is not a separate baseline component, but a deliberate choice of a member of the HIPE family. Other schedules, such as uniform field scales or continuous layer-wise gradients, can also be represented within the same HIPE framework.

The proposed modulation introduces minimal computational overhead. Since $S(\omega)$ is a scalar factor within each frequency subspace, HIPE can be implemented as a lightweight element-wise rescaling applied to the rotated query and key vectors. This operation preserves the standard dot-product attention structure and does not require materializing an additional attention bias matrix.

Consequently, HIPE remains compatible with off-the-shelf memory-efficient attention kernels such as FlashAttention (Dao et al., 2022). In the experiments below, we instantiate this framework with a bipartite layer-wise schedule to keep the design controlled and interpretable.

# 5. Experiments

We design a three-part experimental framework to validate HIPE from complementary perspectives. **Exp 1: mechanism verification** uses controlled synthetic tasks to test whether HIPE facilitates the transition from precise relative positioning to coarse-grained spatial integration. **Exp 2: restricted-budget language modeling** evaluates whether HIPE improves data efficiency under limited pretraining budgets and whether the optimal field scale increases with sequence length. **Exp 3: scaling and sample-limited adaptation** tests whether HIPE remains competitive at a larger pretraining scale and whether its advantages are most pronounced when downstream supervision is limited.

## 5.1. Experimental Setup

**Architecture & Bio-Hierarchy.** We train decoder-only Transformers at two scales (20M and 60M parameters) based on the OLMo architecture (Groeneveld et al., 2024). All models utilize an 8-layer structure. To approximate the hippocampus, we propose a bipartite hierarchical strategy that partitions the model into two functional stages: The first 4 layers form a precision module utilizing standard RoPE to preserve metric details, while the last 4 layers constitute an integration module employing HIPE to enable coarse-grained spatial aggregation. For larger-scale validation, we additionally train 300M-parameter models on 1B C4 tokens and compare RoPE, fixed-$\sigma_p$ HIPE, learnable-$\sigma_p$ HIPE, and XPos (Sun et al., 2023). For sample-limited downstream

*Table 1.* **Task Performance.** HIPE outperforms the standard RoPE in both single-token and block retrieval tasks.

| | Exp 1-1 (One-to-One) | | Exp 1-2 (Block) | |
| Method | Loss | Acc (%) | Loss | Acc (%) |
| --- | --- | --- | --- | --- |
| RoPE | 0.5973 | 81.56 | 0.1155 | 96.32 |
| **HIPE** | **0.3836** | **88.53** | **0.0963** | **97.10** |

adaptation, we fine-tune the resulting 300M pretrained models on SST-2 (Socher et al., 2013) using LoRA (Hu et al., 2022) with varying numbers of labeled examples.

**Training Protocol: Pre-training from Scratch.** To isolate architectural bias from pre-trained knowledge, all models are trained from scratch under fixed compute budgets. We intentionally adopt a data-constrained regime to rigorously evaluate the sample efficiency of the proposed inductive bias: **Exp 1:** Trained for 20k steps on dynamically generated sequences to test algorithmic convergence speed. **Exp 2:** Trained on a restricted budget of 0.1B tokens for WikiText-103 (Merity et al., 2017) and C4 (Raffel et al., 2020). This setting serves to assess real-world data efficiency, verifying whether the hierarchical spatial prior enables the model to generalize effectively without relying on massive-scale pre-training.

## 5.2. Exp 1: Mechanism Verification

**Exp 1-1: One-to-One Associative Recall.** The model must retrieve a specific value $v_q$ given a query key $k_q$ from a long sequence of random key-value pairs. This tests pure addressing over distances. HIPE ($\sigma_p = 200$) outperforms RoPE by a significant margin (88.53% vs. 81.56%, Table 1). The large optimal $\sigma_p$ effectively suppresses relative position cues, encouraging the model to ignore local distance constraints. This is visually confirmed in Appendix E (Figure 10), where HIPE exhibits a broader, more diffusive attention distribution. Unlike RoPE, which may over-attend to local neighborhoods, HIPE's "smearing" effect enables a robust global search mechanism, allowing the model to retrieve keys with higher accuracy.

**Exp 1-2: One-to-Many Block Retrieval.** Each key maps to a continuous block of values ($k_q \rightarrow [v_1, v_2, ..., v_n]$). The model must first locate the key (global search) and then generate the sequence in order (local precision). HIPE achieves the highest accuracy (97.10%) with a moderate field ($\sigma_p = 50$).

## 5.3. Exp 2: Language Modeling under Restricted Pretraining Budgets

Table 2 summarizes the validation perplexity on WikiText-103 and C4. Under the restricted 0.1B-token pretraining budget, HIPE consistently improves over RoPE across the tested 20M/60M configurations, suggesting better data ef-

*Table 2.* **Perplexity comparison on WikiText-103 and C4 datasets.** Green values ($-$) indicate improvement over NoPE, while wine values ($+$) indicate degradation. ($\downarrow$) indicates lower is better.

| Scale | Method | WikiText-103 (PPL $\downarrow$) | | | C4 (PPL $\downarrow$) | | |
|---|---|---|---|---|---|---|---|
| | | L=512 | L=1024 | L=2048 | L=512 | L=1024 | L=2048 |
| **20M** | NoPE | 61.32 | 93.85 | 179.46 | 160.15 | 223.97 | 376.38 |
| | ALiBi | 66.37 +5.0 | **84.74** -9.1 | OOM | 122.58 -37.6 | **168.91** -55.1 | OOM |
| | RoPE | 63.25 +1.9 | 88.30 -5.6 | 133.81 -45.7 | 122.71 -37.4 | 177.30 -46.7 | 276.01 -100.4 |
| | **HIPE (bipartition)** | **59.35** -2.0 | 86.16 -7.7 | **132.97** -46.5 | **117.76** -42.4 | 171.13 -52.8 | **274.93** -101.5 |
| **60M** | NoPE | 50.60 | 69.97 | 152.22 | 123.99 | 201.79 | 355.56 |
| | ALiBi | **42.63** -8.0 | **55.21** -14.8 | OOM | **88.32** -35.7 | **101.62** -100.2 | OOM |
| | RoPE | 48.85 -1.8 | 71.34 +1.4 | 109.17 -43.1 | 99.01 -25.0 | 141.65 -60.1 | 237.90 -117.7 |
| | **HIPE (bipartition)** | 47.98 -2.6 | 68.27 -1.7 | **108.21** -44.0 | 97.54 -26.5 | 130.47 -71.3 | **237.09** -118.5 |

ficiency in small-scale language modeling. HIPE also preserves the computational efficiency of RoPE: unlike ALiBi, whose additive bias requires materializing the full $L \times L$ attention matrix and leads to OOM at $L = 2048$, HIPE remains compatible with memory-efficient attention and scales to longer contexts without additional attention-matrix overhead.

**Mechanism: adaptation to sequence scale.** Our theory predicts that the optimal field scale should increase with the characteristic scale of the environment, since Eq. 13 implies $\sigma_p^{opt}$ grows with $D$. Consistent with this prediction, Figure 1 (panel "HIPE Performance") shows that the best preset $\sigma_p$ shifts to larger values as the sequence length $L$ increases. Thus, shorter sequences favor narrower fields for positional precision, whereas longer contexts benefit from broader fields that integrate distal information.

### 5.4. Exp 3: Scaling Behavior and Sample-Limited Adaptation

**Larger-scale pretraining.** To test whether HIPE remains effective beyond the small-scale setting, we further trained 300M-parameter models on 1B C4 tokens with three random seeds. At this larger scale, HIPE remains competitive with RoPE: RoPE obtains 28.93/31.68 PPL at lengths 512/2048, while learnable-$\sigma_p$ HIPE obtains 28.97/31.69. Fixed-$\sigma_p$ HIPE obtains 29.04/31.93 and outperforms XPos. These results show that HIPE preserves strong pretraining performance at a substantially larger scale, while the learnable-$\sigma_p$ variant narrows the gap to the sharp-field RoPE limit. Together with the low-resource adaptation results below, this supports our view of HIPE as a sample-efficiency-oriented multiscale inductive bias.

**Sample-limited downstream adaptation.** To directly evaluate the sample-efficiency hypothesis, we fine-tune the same 300M pretrained models on SST-2 using LoRA with different numbers of labeled examples. As shown in Table 4, HIPE consistently improves over RoPE in the low-resource

*Table 3.* **Larger-scale C4 pretraining with 300M models and 1B tokens.** HIPE remains competitive with RoPE at larger scale, while learnable-$\sigma_p$ improves over fixed-$\sigma_p$.

| Model | PPL $\downarrow$@L=512 | PPL $\downarrow$@L=2048 |
|---|---|---|
| **RoPE** | **28.93** $\pm$ 0.031 | **31.68** $\pm$ 0.039 |
| HIPE (learnable $\sigma_p$) | 28.97 $\pm$ 0.032 | 31.69 $\pm$ 0.024 |
| HIPE (fixed $\sigma_p$) | 29.04 $\pm$ 0.043 | 31.93 $\pm$ 0.030 |
| XPos | 29.97 $\pm$ 0.055 | 32.75 $\pm$ 0.063 |

regimes from 100 to 2k examples. In contrast, RoPE becomes slightly better when the full 67k-example training set is available. This pattern closely matches the precision–generalization trade-off predicted by our theory: broader multiscale positional bias is beneficial when supervision is limited, whereas the sharp-field RoPE limit can become favorable in data-rich regimes.

*Table 4.* **Low-resource SST-2 adaptation using LoRA.** HIPE improves over RoPE when labeled supervision is limited, while RoPE becomes slightly better in the full-data regime.

| Samples | ACC ($\uparrow$)RoPE@L=512 | ACC ($\uparrow$)HIPE@L=512 |
|---|---|---|
| 100 | 67.3 $\pm$ 5.0 | **68.5** $\pm$ 2.0 |
| 200 | 68.8 $\pm$ 6.4 | **70.1** $\pm$ 5.5 |
| 500 | 75.7 $\pm$ 0.1 | **76.4** $\pm$ 1.6 |
| 1k | 76.3 $\pm$ 0.6 | **77.4** $\pm$ 0.3 |
| 2k | 77.9 $\pm$ 0.3 | **78.2** $\pm$ 1.5 |
| Full | **82.03** $\pm$ 0.40 | 81.77 $\pm$ 0.65 |

### 5.5. Ablation Study

To validate the design of HIPE, we examine its interaction with attention connectivity and compare different $\sigma_p$ scheduling strategies.

**1) Complementarity with local/global attention.** We first compare HIPE with local/global attention under the same 60M, 300M-token C4 setting at context length 1024. As shown in Table 5, local-global attention improves over the RoPE baseline, and adding HIPE achieves the best PPL. This indicates that attention connectivity and positional spec-

*Table 5.* **Complementarity between HIPE and local/global attention** on C4 with a 60M model, 300M tokens, and context length 1024. G and L denote global and local attention, respectively.

| Experiment | Shallow Layers (0–3) | Deep Layers (4–7) | PPL($\downarrow$) |
|---|---|---|---|
| Baseline | global attention+RoPE | global attention+RoPE | 62.82 |
| Local-Global | Local attention+RoPE | global attention+RoPE | 59.36 |
| Global-Local | global attention+RoPE | Local attention+RoPE | 64.32 |
| Bipartite HIPE | global attention+RoPE | global attention+HIPE, $\sigma_p = 500$ | 62.71 |
| Global-Local HIPE | global attention+RoPE | Local attention+HIPE, $\sigma_p = 500$ | 64.30 |
| **Local-Global HIPE** | Local attention+RoPE | global attention+HIPE, $\sigma_p = 500$ | **59.01** |

tral shaping provide complementary benefits, supporting our sharp-to-broad hierarchy.

**2) Necessity of the sharp-to-broad order.** Table 6 further studies $\sigma_p$ scheduling on the 20M model. Uniform schedules ($\sigma_p = 50$ or $200$ across all layers) fail to match RoPE, suggesting that a single spatial scale cannot simultaneously support fine positional discrimination and broad integration. The reverse schedule, which places broad fields in shallow layers and sharp fields in deeper layers, also degrades performance.

**3) Discrete hierarchy vs. continuous gradient.** We also test continuous-gradient schedules where $\sigma_p$ increases gradually across deeper layers. Although biologically plausible, these smoother transitions underperform the discrete bipartite schedule in our setting, suggesting that the simple sharp-to-broad partition is a more effective controlled instantiation for the current 8-layer, low-data regime.

*Table 6.* **Ablation study on $\sigma_p$ scheduling strategies (WikiText-103, 20M, L=2048).** We investigate the optimal structural design by comparing: (1) Uniform vs. Hierarchical, and (2) Continuous vs. Discrete transitions. "Continuous Gradient" (CG) refers to increasing $\sigma_p$ linearly across the deep layers: CG-1 ($\sigma_p \in \{50, 200, 500, 700, 1000\}$ from layer 4), CG-2 ($\sigma_p \in \{50, 200, 500, 700\}$ from layer 5), and CG-3 ($\sigma_p \in \{200, 700\}$ from layer 7).

| Method / Configuration | PPL ($\downarrow$) |
|---|---|
| RoPE | 133.81 |
| Uniform ($\sigma_p = 50$) | 138.45 +4.64 |
| Uniform ($\sigma_p = 200$) | 134.10 +0.29 |
| Reverse Gradient (broad $\rightarrow$ sharp) | 136.22 +2.41 |
| CG-1 (early start, layers 4-8) | 136.37 +2.56 |
| CG-2 (mid start, layers 5-8) | 135.96 +2.15 |
| CG-3 (late start, layers 7-8) | 135.12 +1.31 |
| **HIPE (bipartition)** | **132.97 -0.84** |

## 6. Conclusion

This work establishes a unified framework linking the anatomical organization of the hippocampus to the computational principles of general intelligence. We demonstrated that the dorsal-ventral gradient of place field sizes is not merely an anatomical curiosity but a functional necessity, managing a fundamental trade-off between high-fidelity memorization and few-shot generalization. By mathematically deriving how grid-to-place projections determine field scale, we showed that a multi-scale population code ensures robust learning without prior knowledge of the task's spatial structure. Translating this biological inductive bias into Transformers, we introduced HIPE. This mechanism enabl a phase transition from rigid positional arithmetic to flexible semantic addressing and significantly improving sample efficiency in language modeling. Ultimately, our findings suggest that the brain's strategy of distributing representations across diverse scales provides a blueprint for designing more adaptive and data-efficient AI systems.

## Limitations

Biological simplifications: The theoretical model primarily employs linear transformations and assumes idealized Gaussian place fields. Future work should explore the impact of biologically plausible non-linearities (e.g., firing thresholds, dendritic computations) and the influence of environmental boundaries beyond the periodic conditions assumed for analytical tractability.

Static vs. dynamic coding: Our current model is static. A more complete picture of hippocampal function would require integrating dynamic temporal aspects, such as theta oscillations, phase precession, and synaptic plasticity mechanisms (e.g., Hebbian rules that could dynamically sculpt the frequency-dependent weight decay).

Beyond space: Finally, while HIPE successfully applies spatial principles to language, further research is needed to conceptualize how this "cognitive space" maps onto other non-spatial, relational domains. Extending the theory to account for these abstract eigenmodes presents a fascinating direction for future inquiry.

## Impact Statement

This paper introduces a bio-inspired inductive bias that improves the sample efficiency and generalization of Trans-

former models. Our approach allows models to achieve competitive performance with significantly less training data. This contributes to the broader goal of sustainable AI by lowering the energy footprint of model training and democratizing access to powerful neural networks in data-constrained environments.

## Acknowledgement

This work was supported in part by the Beijing Major Science and Technology Project under Contract no. Z251100008125055. This work was supported by Beijing Academy of Artificial Intelligence (BAAI). This work was also supported by the National Natural Science Foundation of China (NSFC) under Grant No. 62576011. The authors would like to thank the useful discussions with Chun Xu and Shou Qiu on hippocampus connections.

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

## A. Non-linear Activation Function

We implemented a non-linear readout in the same task shown in Figure 3, $\hat{y} = \tanh(\mathbf{W} \cdot \text{concat}(\mathbf{s}, \mathbf{p}(x)))$, evaluated with 100 place cells (learning rate 0.01, 300 epochs). As shown below, the optimal place field size for few-shot learning persists: $\sigma_p \approx 15$ consistently yields the highest accuracy across all tested sample sizes $S$. This confirms the observed trade-off is a fundamental property of the representational spectral structure, not an artifact of linear readouts. While expressive non-linear readouts may achieve perfect accuracy given infinite data, the sample efficiency advantage of optimally-sized place fields remains robust in the few-shot regime.

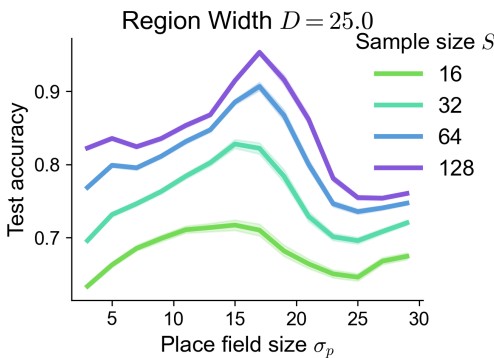

*Figure 8.* Relationship between test accuracy and place field width under non-linear activation function

## B. Derivation of the Generalization Error

The derivation of the average generalization error for kernel regression presented here follows the framework established by Bordelon & Pehlevan (2022), which builds upon work by Canatar et al. (2021). We adapt the notation to match the main paper.

The average generalization error $E_g$ is defined as the expected squared difference between the target function $y(x)$ and the learned function $f(x) = \mathbf{w} \cdot \mathbf{p}(x)$, averaged over the data distribution and training sets $\mathcal{D}_S$ of size $S$:

$$E_g = \left\langle \|f(x; \mathcal{D}_S) - y(x)\|_2^2 \right\rangle_{x, \mathcal{D}_S}. \tag{20}$$

The functions $y(x)$ and $f(x)$ can be expanded in the eigenbasis $\{\psi_k(x)\}$ of the covariance kernel $\Sigma(x, x') = \mathbf{p}(x)^\top \mathbf{p}(x')$. Let $\lambda_k$ be the eigenvalues corresponding to $\psi_k(x)$.

$$y(x) = \sum_k \nu_k \psi_k(x) \tag{21}$$

$$f(x; \mathcal{D}_S) = \sum_k \hat{\nu}_k(\mathcal{D}_S) \psi_k(x) \tag{22}$$

Due to the orthonormality of the eigenfunctions, the generalization error is:

$$E_g = \sum_k \langle (\hat{\nu}_k(\mathcal{D}_S) - \nu_k)^2 \rangle_{\mathcal{D}_S}. \tag{23}$$

The coefficients $\hat{\nu}_k$ are learned via kernel ridge regression, minimizing an empirical loss with L2 regularization term $\alpha \sum_k \hat{\nu}_k^2 / \lambda_k$. The optimal coefficients for a given dataset $\mathcal{D}_S = \{(x^\mu, y^\mu)\}_{\mu=1}^S$ are denoted $\hat{\boldsymbol{\nu}}(\mathcal{D}_S)$. The deviation from the true coefficients $\boldsymbol{\nu}$ can be expressed as:

$$\hat{\boldsymbol{\nu}}(\mathcal{D}_S) - \boldsymbol{\nu} = -\alpha \left( \boldsymbol{\Psi}\boldsymbol{\Psi}^\top + \alpha\boldsymbol{\Lambda}^{-1} \right)^{-1} \boldsymbol{\Lambda}^{-1}\boldsymbol{\nu}, \tag{24}$$

where $\boldsymbol{\Psi}$ is a matrix with entries $\Psi_{k\mu} = \psi_k(x^\mu)$, and $\boldsymbol{\Lambda}$ is a diagonal matrix of eigenvalues $\lambda_k$. The generalization error for a specific dataset $\mathcal{D}_S$ is then:

$$E_g(\mathcal{D}_S) = \|\hat{\boldsymbol{\nu}}(\mathcal{D}_S) - \boldsymbol{\nu}\|^2$$
$$= \alpha^2 \boldsymbol{\nu}^\top \boldsymbol{\Lambda}^{-1} \left( \boldsymbol{\Psi}\boldsymbol{\Psi}^\top + \alpha\boldsymbol{\Lambda}^{-1} \right)^{-2} \boldsymbol{\Lambda}^{-1}\boldsymbol{\nu}. \tag{25}$$

Let $\mathbf{G}(\mathcal{D}_S) = \left(\frac{1}{\alpha}\boldsymbol{\Psi}\boldsymbol{\Psi}^\top + \boldsymbol{\Lambda}^{-1}\right)^{-1}$. Then, $E_g(\mathcal{D}_S) = \boldsymbol{\nu}^\top \boldsymbol{\Lambda}^{-1}\mathbf{G}(\mathcal{D}_S)^2\boldsymbol{\Lambda}^{-1}\boldsymbol{\nu}$. Averaging $E_g(\mathcal{D}_S)$ over all datasets $\mathcal{D}_S$ can be done using techniques from statistical mechanics of disordered systems (e.g., replica theory or dynamical mean-field theory). This involves averaging $\mathbf{G}(\mathcal{D}_S)$. The averaged matrix $\langle\mathbf{G}(\mathcal{D}_S)\rangle_{\mathcal{D}_S}$, denoted $\mathbf{G}(S)$, can be found by solving a set of self-consistent equations. The derivation involves introducing an auxiliary variable $J$ and defining $\mathbf{G}(S,J)_{k,l} = \langle(\frac{1}{\alpha}\boldsymbol{\Psi}\boldsymbol{\Psi}^\top + \boldsymbol{\Lambda}^{-1} + J\mathbf{I})^{-1}_{k,l}\rangle_{\mathcal{D}_S}$. The solution for the diagonal elements $G_k(S,J)$ is:

$$G_k(S,J) = \left(\frac{S}{\kappa(S,J)} + J + \lambda_k^{-1}\right)^{-1}$$
$$= \frac{\kappa(S,J)\lambda_k}{\lambda_k S + \kappa(S,J) + J\kappa(S,J)\lambda_k}, \tag{26}$$

where $\kappa(S,J)$ is a scalar quantity determined self-consistently by:

$$\kappa(S,J) = \alpha + \sum_k G_k(S,J). \tag{27}$$

Setting $J = 0$, this gives Eqn. 8 from the main paper:

$$\kappa = \alpha + \kappa\sum_k \frac{\lambda_k}{\lambda_k S + \kappa}, \tag{28}$$

where $\kappa \equiv \kappa(S,0)$. The average generalization error is related to the derivative of $\mathbf{G}(S,J)$ with respect to $J$:

$$\langle\mathbf{G}(\mathcal{D}_S,J)^2\rangle_{\mathcal{D}_S} = -\frac{\partial}{\partial J}\mathbf{G}(S,J). \tag{29}$$

So, at $J = 0$:

$$E_g = \sum_k \frac{\nu_k^2}{\lambda_k^2}\left(-\frac{\partial G_k(S,J)}{\partial J}\Big|_{J=0}\right). \tag{30}$$

We have $-\frac{\partial G_k(S,J)}{\partial J} = G_k(S,J)^2\left(1 - \frac{S}{\kappa(S,J)^2}\frac{\partial\kappa(S,J)}{\partial J}\right)$. At $J = 0$, $G_k(S,0) = \frac{\kappa\lambda_k}{\lambda_k S + \kappa}$. The term $\frac{\partial\kappa(S,J)}{\partial J}\big|_{J=0}$ can be found from $\kappa(S,J) = \alpha + \sum_k G_k(S,J)$:

$$\frac{\partial\kappa}{\partial J} = \sum_k \frac{\partial G_k}{\partial J} = \sum_k -G_k^2\left(1 - \frac{S}{\kappa^2}\frac{\partial\kappa}{\partial J}\right). \tag{31}$$

Solving for $\frac{\partial\kappa}{\partial J}\big|_{J=0}$:

$$\frac{\partial\kappa}{\partial J}\Big|_{J=0} = \frac{-\sum_k G_k(S,0)^2}{1 - \frac{S}{\kappa^2}\sum_k G_k(S,0)^2} = \frac{-\kappa^2\sum_k \frac{\lambda_k^2}{(\lambda_k S + \kappa)^2}}{1 - S\sum_k \frac{\lambda_k^2}{(\lambda_k S + \kappa)^2}}. \tag{32}$$

Let $\gamma = S\sum_k \frac{\lambda_k^2}{(\lambda_k S + \kappa)^2}$. Then $\left(1 - \frac{S}{\kappa^2}\frac{\partial\kappa}{\partial J}\big|_{J=0}\right) = 1 - \frac{S}{\kappa^2}\frac{-\kappa^2(\gamma/S)}{1-\gamma} = 1 + \frac{\gamma}{1-\gamma} = \frac{1}{1-\gamma}$. Substituting this back into the expression for $E_g$:

$$E_g = \sum_k \frac{\nu_k^2}{\lambda_k^2}G_k(S,0)^2\frac{1}{1-\gamma}$$
$$= \sum_k \frac{\nu_k^2}{\lambda_k^2}\frac{\kappa^2\lambda_k^2}{(\lambda_k S + \kappa)^2}\frac{1}{1-\gamma}. \tag{33}$$

This yields:

$$E_g = \frac{\kappa^2}{1-\gamma}\sum_k \frac{\nu_k^2}{(\lambda_k S + \kappa)^2}, \tag{34}$$

with $\kappa = \alpha + \kappa\sum_k \frac{\lambda_k}{\lambda_k S + \kappa}$ and $\gamma = S\sum_k \frac{\lambda_k^2}{(\lambda_k S + \kappa)^2}$.

## C. Derivation of Optimal Place Field Size

### C.1. Grid Codes as Eigenmodes of Place Codes

In order to derive the eigenvalues of place code (Eq.10), we first prove that grid codes are the eigenmodes of the spatial covariance function $\Sigma(x, x') = \mathbf{p}(x)^\top \mathbf{p}(x)$. In a 1D periodic environment of length $L$, we consider the spatial covariance function $\Sigma(x, x')$. Due to translational invariance, we can write the covariance as a function of the relative displacement: $\Sigma(x, x') = \Sigma(x - x')$.

To find the eigenmodes of the place code, we solve the integral equation:

$$\int_0^L \Sigma(x - x')\phi(x')dx' = \lambda\phi(x) \tag{35}$$

**Proof.** We hypothesize that the eigenfunctions $\phi(x)$ are the Fourier basis functions:

$$\phi_k(x) = e^{ikx}, \quad \text{where } k = \frac{2\pi n}{L}, n \in \mathbb{Z} \tag{36}$$

Substituting $\phi_k(x)$ into the left-hand side (LHS) of Eq. 35:

$$\text{LHS} = \int_0^L \Sigma(x - x')e^{ikx'} dx' \tag{37}$$

We perform a change of variables. Let $u = x - x'$, which implies $x' = x - u$ and $dx' = -du$. Since the domain is periodic, the limits of integration over one period remain $[0, L]$:

$$\text{LHS} = \int_0^L \Sigma(u)e^{ik(x-u)} du \tag{38}$$

$$= e^{ikx} \int_0^L \Sigma(u)e^{-iku} du \tag{39}$$

Notice that the integral term is independent of $x$. This term corresponds to the eigenvalue $\lambda_k$, which is the Fourier transform of the covariance kernel at spatial frequency $k$:

$$\lambda_k = \int_0^L \Sigma(u)e^{-iku} du \tag{40}$$

Thus, we obtain:

$$\int_0^L \Sigma(x - x')\phi_k(x')dx' = \lambda_k\phi_k(x) \tag{41}$$

This confirms that the periodic oscillations—the 1D analogues of grid codes—are the eigenmodes of the translationally invariant place cell covariance.

### C.2. Eigenvalues of the Covariance Matrix

Starting from Equation 2, the covariance function of the place cell population activity can be written as:

$$\Sigma(x, x') = [\boldsymbol{W}\boldsymbol{g}(x)]^\top [\boldsymbol{W}\boldsymbol{g}(x')]$$
$$= \boldsymbol{g}(x)^\top \boldsymbol{W}^\top \boldsymbol{W} \boldsymbol{g}(x'), \tag{42}$$

where $\boldsymbol{W}_{ij}$ denotes the connection weight from grid cell $j$ to place cell $i$.

To generate the desired Gaussian place fields, and assuming that the connection weights between grid cells and place cells are independent across place cells, we have, the weight matrix $\boldsymbol{W}$ must satisfy:

$$W_{i,2k-1}W_{j,2k-1} = C^2 \, \delta_{ij} \, \exp\left[-\sigma_p^2 k^2 \left(\frac{2\pi}{L}\right)^2\right], \tag{43}$$

where $C$ is a constant and $\delta_{ij}$ is the Kronecker delta.

Therefore, the matrix $\boldsymbol{W}^\top \boldsymbol{W}$ takes the diagonal form:

$$\boldsymbol{W}^\top \boldsymbol{W} = C^2 \operatorname{diag}\big\{ e^{-\sigma_p^2 (\frac{2\pi}{L})^2}, \, e^{-\sigma_p^2 (\frac{2\pi}{L})^2}, \, e^{-4\sigma_p^2 (\frac{2\pi}{L})^2},$$
$$e^{-4\sigma_p^2 (\frac{2\pi}{L})^2}, \, \ldots, \, e^{-k^2 \sigma_p^2 (\frac{2\pi}{L})^2}, \, e^{-k^2 \sigma_p^2 (\frac{2\pi}{L})^2}, \, \ldots \big\}$$
$$= C^2 \operatorname{diag}\{\lambda_1, \, \lambda_1, \, \lambda_2, \, \lambda_2, \, \ldots, \, \lambda_k, \, \lambda_k, \, \ldots \}, \tag{44}$$

where $\lambda_k$ denotes the eigenvalue associated with the $k$-th spatial frequency mode.

Since the Fourier basis functions $\boldsymbol{g}(x)$ are the eigenfunctions of $\boldsymbol{\Sigma}$ (Appendix C.1), the eigenvalues $\lambda_k$ are given by:

$$\lambda_k \propto \exp\left[ -\sigma_p^2 k^2 \left( \frac{2\pi}{L} \right)^2 \right].$$

## C.3. Optimal Place Field Size under Few-Shot Learning Regime

Substituting $\lambda_k = \exp\left[ -\sigma_p^2 k^2 \left( \frac{2\pi}{L} \right)^2 \right]$ into Equation 8 with $\alpha = 0$ (i.e., no L2-regularization), we have:

$$2 \sum_k \frac{e^{-\sigma_p^2 k^2 (2\pi/L)^2}}{e^{-\sigma_p^2 k^2 (2\pi/L)^2} S + \kappa} = 1. \tag{45}$$

Where the factor 2 is due to the degeneracy of cosine and sine Fourier modes. Approximating the sum as an integral:

$$\sum_k \frac{\lambda_k}{\lambda_k S + \kappa} \approx \frac{L}{2\pi \sigma_p \kappa} \int_0^\infty \frac{e^{-u^2}}{1 + \frac{S}{\kappa} e^{-u^2}} \, du. \tag{46}$$

When $S \ll \kappa$, this can be approximated as:

$$\approx \frac{L}{2\pi \sigma_p \kappa} \int_0^\infty e^{-u^2} \, du$$
$$= \frac{L}{2\sqrt{2\pi} \sigma_p \kappa}.$$

Thus, we derive:

$$\kappa = \frac{L}{\sqrt{2\pi} \sigma_p}. \tag{47}$$

For the task setting depicted in Figure 3, an analysis of the task's spectral content (see Appendix C.4 for details) reveals that the projection coefficients $\nu_k$ exhibit a pronounced peak. This dominant contribution occurs at a spatial frequency mode $n \approx L/(2D)$, corresponding to the characteristic scale $D$ of the spatial regions in the task. To gain analytical insight into how the generalization error $E_g$ and the optimal place field size $\sigma_p$ depend on this primary task scale $D$, we simplify our analysis by focusing on the influence of this dominant mode. We approximate the task spectrum as being entirely concentrated at this single mode $n$, i.e., $\nu_k = \nu_n \delta_{k,n}$ (where $\nu_n$ represents the magnitude of the projection onto this dominant mode, and $\delta_{k,n}$ is the Kronecker delta). While this is a simplification of the full task spectrum, it allows for a tractable derivation of the key scaling relationships, the validity of which is supported by our comprehensive numerical simulations (e.g., Figure 6) that use the complete task structure. Substituting this approximation, $\nu_k = \nu_n \delta_{k,n}$, into the expression for $E_g$ (Equation 7) yields:

$$E_g = \frac{\kappa^2}{1 - \gamma} \frac{\nu_n^2}{(e^{-\sigma_p^2 n^2 (2\pi/L)^2} S + \kappa)^2} \tag{48}$$
$$= \frac{\nu_n^2}{1 - \gamma} \frac{1}{(C \sigma_p e^{-\sigma_p^2 n^2 (2\pi/L)^2} + 1)^2}, \tag{49}$$

where $C = \frac{\sqrt{2\pi} S}{L}$. Therefore, we can find the optimal place field size by minimizing $E_g$, and this corresponds to maximizing $\sigma_p e^{-\sigma_p^2 n^2 (2\pi/L)^2}$ with respect to $\sigma_p$. Thus we have:

$$\sigma_p^{\mathrm{opt}} = \frac{L}{2\sqrt{2\pi} n} \sim L \cdot n^{-1} \sim D. \tag{50}$$

### C.4. Spectral Content of the Task and Single-Mode Approximation

In Section 3.3 of the main paper, we simplified the analysis of the generalization error $E_g$ by assuming that the task's spectral content is dominated by a single eigenmode $n$. This approximation, $\nu_k = \nu_n \delta_{k,n}$ (where $\nu_k$ are the projection coefficients of the target function $y(x)$ onto the eigenmodes $\psi_k(x)$ of the population code kernel), is based on the spectral characteristics of the context-dependent computation task described in Figure 3 of the main paper.

The target function $y(x)$ for this task is a piecewise constant function, alternating between two values depending on the spatial region. For example, if the task is to output $y_1$ in regions where an 'AND' operation is required and $y_2$ in regions where an 'OR' operation is required, and these regions have a characteristic width $D$, the target function $y(x)$ will resemble a square wave or a series of square pulses with period $2D$ (or fundamental wavelength related to $D$).

When such a function is decomposed into its Fourier components (which are the eigenmodes $\psi_k(x)$ in our translationally invariant system), the power spectrum (i.e., $\nu_k^2$) typically exhibits a dominant peak at the frequency corresponding to the fundamental period of the target function. Higher harmonics will be present, but their amplitudes often decay.

Figure 9 illustrates the squared projection coefficients $\nu_k^2$ for a representative instance of the task structure used in our simulations. The total length of the environment is $L = 100$ and the region width is $D = 25$. In this case, the task function $y(x)$ changes its value every $D = 25$ units of space, implying a fundamental spatial period of $2D = 50$. The corresponding dominant spatial frequency mode $n$ would be $n = L/(2D) = 100/50 = 2$.

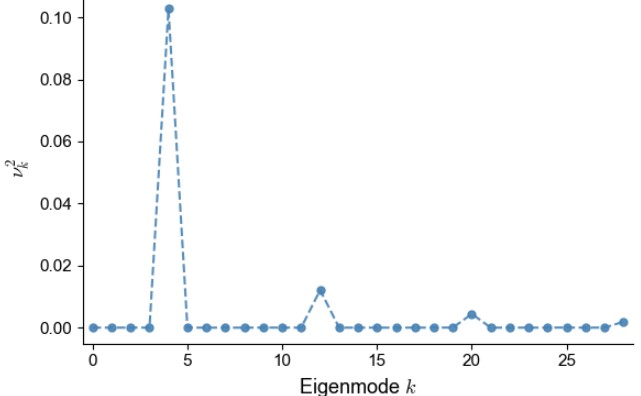

*Figure 9.* Squared projection coefficients $\nu_k^2$ of the target function $y(x)$ onto the eigenmodes $g_k(x)$ for the context-dependent task (environment length $L = 100$, region width $D = 25$). The plot shows a dominant peak at eigenmode $g_4(x) = \sin(\frac{4\pi x}{L})$, corresponding to the fundamental spatial frequency of the task structure ($n \approx L/(2D)$). The coefficients for other modes are significantly smaller.

As seen in Figure 9, the $\nu_k^2$ values show a pronounced peak at $k = 4$. The contributions from other eigenmodes are substantially smaller. This observation justifies the approximation that the task is primarily aligned with a single eigenmode. While other modes contribute, the analytical tractability gained by focusing on the dominant mode allows us to derive the key scaling relationships for the optimal place field size, as presented in the main paper. The numerical simulations (e.g., Figure 5 in the main paper) which use the full task structure confirm the validity of the insights derived from this single-mode approximation.

## D. Detailed Formulation of Rotary Positional Embeddings

In this section, we provide the complete mathematical formulation of the standard rotary positional embedding (RoPE) and demonstrate that it corresponds to the "dorsal limit" of our proposed HIPE framework.

### D.1. Standard RoPE Formulation

Given an input vector $\mathbf{x} \in \mathbb{R}^d$ and a position index $m$, RoPE applies a rotation operation in $\mathbb{R}^d$. The dimension $d$ is assumed to be even. The vector is divided into $d/2$ pairs of coordinates. For each subspace $j \in \{1, \ldots, d/2\}$, the rotation is defined by a specific frequency $\omega_j$.

The canonical choice for frequencies in RoPE follows a geometric progression:

$$\omega_j = \theta_{\text{base}}^{-2(j-1)/d}, \quad \text{typically } \theta_{\text{base}} = 10000. \tag{51}$$

The full rotation matrix $\mathbf{R}_{\Theta,m}$ is a block-diagonal matrix:

$$\mathbf{R}_{\Theta,m} = \begin{pmatrix} \mathbf{M}_1 & \mathbf{0} & \cdots & \mathbf{0} \\ \mathbf{0} & \mathbf{M}_2 & \cdots & \mathbf{0} \\ \vdots & \vdots & \ddots & \vdots \\ \mathbf{0} & \mathbf{0} & \cdots & \mathbf{M}_{d/2} \end{pmatrix}, \quad \text{where } \mathbf{M}_j = \begin{pmatrix} \cos m\omega_j & -\sin m\omega_j \\ \sin m\omega_j & \cos m\omega_j \end{pmatrix}. \tag{52}$$

Applying this rotation to the query $\mathbf{q}$ and key $\mathbf{k}$ vectors allows the inner product to encode relative position. By viewing each pair of elements as a complex number $q_j, k_j \in \mathbb{C}$, the inner product simplifies to:

$$\langle f(\mathbf{q}, m), f(\mathbf{k}, n) \rangle = \text{Re}\left[ \sum_{j=1}^{d/2} q_j k_j^* e^{i\omega_j(m-n)} \right]. \tag{53}$$

## D.2. RoPE as the Dorsal Limit

In our HIPE framework, the spatial attention kernel is expressed as a spectral synthesis over rotary frequency components. For a general frequency-dependent modulation, the positional component of the attention score can be written as

$$\text{Attn}_{\text{HIPE}}(\Delta) \propto \sum_{j=1}^{d/2} S(\omega_j)^2 \cos(\omega_j \Delta). \tag{54}$$

Standard RoPE applies no additional scaling to the rotary basis frequencies. Equivalently, it uses a uniform discrete spectral weight over the RoPE frequency index,

$$S_{\text{RoPE}}(\omega_j) \equiv 1, \qquad j = 1, \ldots, d/2. \tag{55}$$

This unscaled spectrum should be distinguished from the density-compensated Gaussian construction used by the nonzero-field HIPE branch. RoPE itself does not require a Jacobian compensation term, because it is defined directly as an unweighted discrete sum over the prescribed log-spaced RoPE frequencies. The $\omega_j$-dependent compensation factor in HIPE is introduced only when these log-spaced frequencies are reinterpreted as quadrature points for approximating a continuous Gaussian Fourier kernel.

**1. Theoretical Limit.** To analyze the asymptotic behavior of the unscaled RoPE branch, we first consider an idealized continuous-frequency limit in which the frequency basis becomes dense over a symmetric frequency domain. Noting that $\cos(\omega\Delta) = \text{Re}[e^{i\omega\Delta}]$, a constant spectrum over the continuous frequency measure yields

$$\int_{-\infty}^{\infty} e^{i\omega\Delta}\, \mathrm{d}\omega = 2\pi\delta(\Delta). \tag{56}$$

Thus, in the distributional sense, a flat continuous spectrum corresponds to a Dirac delta kernel, representing the limiting case of maximal spatial locality.

**2. Finite-Dimensional Reality.** In practice, the model dimension $d$ is finite and the available frequency range is bounded. In the idealized case of a uniform spectrum truncated to $[-\Omega_{\max}, \Omega_{\max}]$, the inverse Fourier transform becomes

$$\int_{-\Omega_{\max}}^{\Omega_{\max}} e^{i\omega\Delta}\, \mathrm{d}\omega = 2\Omega_{\max}\, \text{sinc}(\Omega_{\max}\Delta), \tag{57}$$

where $\text{sinc}(x) = \sin(x)/x$. This finite-bandwidth kernel is a band-limited approximation to a delta function. Although practical RoPE uses a finite set of log-spaced frequencies rather than a uniformly sampled frequency grid, its unscaled discrete spectrum preserves the same sharp-field intuition: all available rotary frequency components contribute without additional attenuation.

**3. Connection to HIPE.** The nonzero-$\sigma_p$ HIPE branch aims to synthesize a broader Gaussian-like positional field,

$$\exp\left(-\frac{\Delta^2}{2\sigma_p^2}\right), \tag{58}$$

by applying frequency-dependent spectral modulation to the same rotary basis. When RoPE frequencies are treated as log-spaced quadrature points for a continuous Fourier integral, their frequency density satisfies $\rho(\omega) \propto 1/\omega$. Therefore, the density-compensated HIPE branch uses spectral weights satisfying

$$S_{\mathrm{HIPE}}(\omega; \sigma_p)^2 \rho(\omega) \propto \exp\left(-\frac{\sigma_p^2 \omega^2}{2}\right), \tag{59}$$

so that the effective spectrum with respect to the uniform-frequency measure has the desired Gaussian form. In the limit $\sigma_p \to 0$, this effective continuous-frequency spectrum becomes flat and the induced kernel approaches a Dirac delta in the distributional sense:

$$\lim_{\sigma_p \to 0} \frac{1}{\sqrt{2\pi}\sigma_p} \exp\left(-\frac{\Delta^2}{2\sigma_p^2}\right) = \delta(\Delta). \tag{60}$$

**Conclusion.** RoPE and HIPE therefore share a unified sharp-to-broad geometric interpretation, but they should be distinguished at the level of their discrete spectral weights:

- **RoPE** is the unscaled discrete sharp-field branch, defined by $S_{\mathrm{RoPE}}(\omega_j) = 1$ over the RoPE frequency index. It represents the dorsal-limit reference with maximal available positional precision.

- **HIPE** uses a nonzero-$\sigma_p$, density-compensated Gaussian modulation to synthesize broader place-like kernels. As $\sigma_p \to 0$, this branch approaches a sharp delta-like kernel in the continuous-frequency approximation, but it does not imply element-wise equality $S_{\mathrm{HIPE}}(\omega_j; \sigma_p) \to S_{\mathrm{RoPE}}(\omega_j)$ on the discrete RoPE grid.

Thus, standard RoPE represents the unscaled dorsal-limit reference of the HIPE hierarchy, whereas nonzero-$\sigma_p$ HIPE corresponds to broader positional integration analogous to larger hippocampal place fields.

## E. Visualization of Attention Patterns

To validate the impact of HIPE on the model's internal mechanism, we visualize the evolution of attention weights across layers in the Associative Recall task (Exp 1-1). Figure 10 compares the attention maps of the 20M models for RoPE and HIPE.

In the shallow layers, both models exhibit similar diagonal patterns, focusing on local neighbors. However, a significant divergence emerges in the deep layers:

- **Standard RoPE** retains a sharp diagonal concentration. This confirms that it is constrained by **rigid relative positioning**, restricting the model to a local effective receptive field.

- **HIPE** displays a visibly broader and more diffusive attention distribution. The Gaussian decay effectively "smears" the positional bias, suppressing local distance constraints. This enables **global spatial integration**, allowing the attention head to ignore relative distance and retrieve specific key-value pairs from the global context.

## F. Related Works

### F.1. Neuroscience

While previous computational studies focus on the emergence of grid-like representations (Cueva & Wei, 2018; Banino et al., 2018; Sorscher et al., 2023), our work addresses the structural origin of multiscale place fields from grid cell projections. Previous work (Solstad et al., 2006) employed a continuous, logarithmically sampled distribution of grid scales, whereas our model adopts discrete grid modules, consistent with the experimental finding that grid scales are organized into distinct

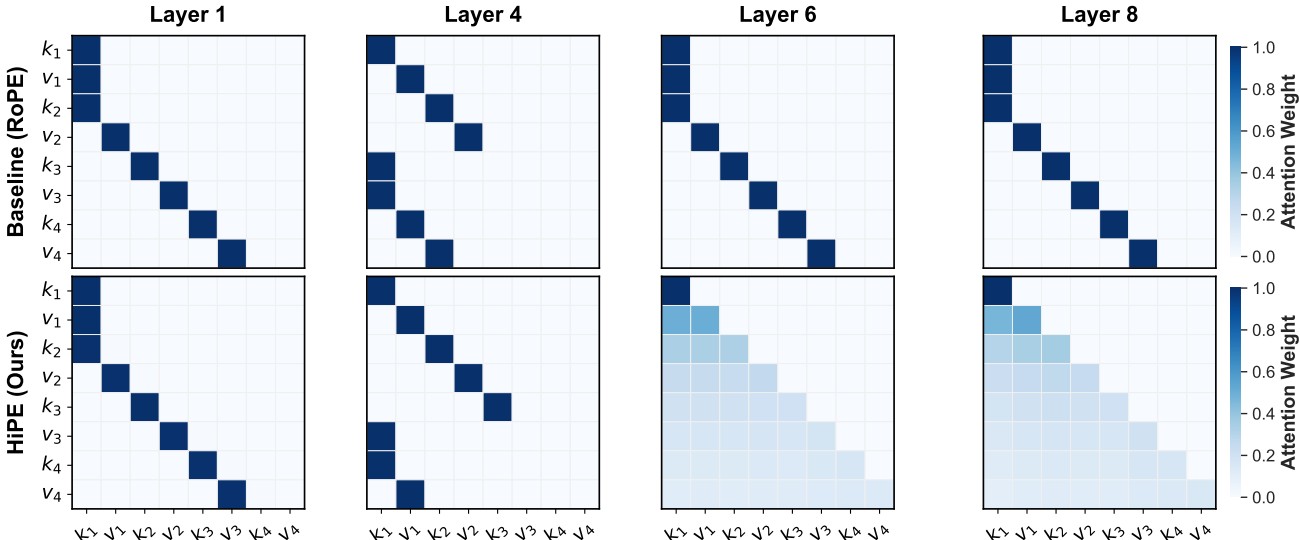

*Figure 10.* **Evolution of Attention Patterns (Exp 1-1).** Comparison of attention weights from Layer 1 to Layer 8. **Top (RoPE):** Maintains strong diagonal concentration throughout all layers, indicating a reliance on **rigid relative positioning** (local metric cues). **Bottom (HIPE):** Evolves from local precision in shallow layers to broad coverage in deep layers (Layer 8). The diffusive pattern visually confirms the transition to **global spatial integration**, where the model suppresses distance penalties to access distant tokens.

discrete bands (Stensola et al., 2012). We move beyond observation to demonstrate that this biological organization serves as a functional inductive bias for artificial intelligence. Unlike prior works that suggest high-level links between the hippocampus and Transformers (Whittington et al., 2022; Gershman et al., 2025), we derive a formal mathematical mapping between grid-place cell scales and Transformer positional embeddings. This enables a principled, bio-inspired mechanism—HIPE—that directly improves sample efficiency in practical learning systems.

## F.2. Positional Encodings in Transformers

### F.2.1. The Evolution of Positional Representations

The mechanism for encoding sequence order has evolved significantly. Original Transformers employed absolute sinusoidal positional embeddings (APE) (Vaswani et al., 2017), later adapted into learnable forms (Gehring et al., 2017; Lan et al., 2020). However, APEs generally fail to generalize to sequences longer than those seen during training. This limitation catalyzed the shift toward Relative Positional Embeddings (RPE) (Shaw et al., 2018), which encode token distances directly into the attention mechanism. Prominent RPE variants include the recurrence-based Transformer-XL (Dai et al., 2019), the disentangled attention of DeBERTa (He et al., 2021), and the learnable relative bias of T5 (Raffel et al., 2020).

Rotary Positional Embedding (RoPE) (Su et al., 2024) unifies absolute and relative approaches by encoding position as a rotation in the complex plane. Due to its mathematical elegance and superior performance, RoPE has been widely adopted in foundation models like Llama and PaLM (Touvron et al., 2023; Chowdhery et al., 2023). In our framework, we theoretically ground RoPE not merely as an engineering solution, but as the dorsal limit of the hippocampal code ($\sigma_p \to 0$), corresponding to a high-frequency grid module with no spatial blurring.

### F.2.2. Context Scaling and Inductive Biases

A central challenge in current LLM research is adapting positional codes for varying context lengths and resolutions.

**Length Extrapolation via Scaling.** Numerous methods modulate the RoPE spectrum to enable length extrapolation. Linear and NTK-aware scaling (Rozière et al., 2024) interpolate position indices to prevent resolution collapse. More advanced techniques like YaRN (Peng et al., 2024) and XPos (Sun et al., 2023) introduce frequency-dependent scaling or exponential decay to stabilize attention over long distances. Crucially, while XPos employs decay to stabilize long-term dependencies for extrapolation, our HIPE framework derives its spectral decay $S(\omega)$ from the biological principle of place field expansion, specifically to optimize sample efficiency and few-shot learning rather than infinite context length.

**Locality Biases.** Parallel to spectral manipulation, other works enforce locality through additive biases. ALiBi (Press et al.,

2022) applies a static linear penalty to attention scores, and Sandwich (Chi et al., 2023) combines sinusoidal and learned components to improve theoretical expressivity. While ALiBi achieves strong extrapolation, its rigid bias structure penalizes the global semantic retrieval required for complex reasoning. HIPE offers a middle ground: a "soft," tunable spectral filter that favors locality (ventral-like broad fields) without permanently suppressing distant information, preserving the flexibility inherent in the attention mechanism.

