# OpenReview forum: "The Hippocampal Place Field Gradient: A Bio-inspired Framework Building Multiscale Representation for Better Sample Efficiency"
_ICML.cc/2026/Conference — ICML 2026 regular_

### Official Review · Reviewer_n6qN · 2026-03-10

**Soundness:** 3
**Presentation:** 2
**Significance:** 2
**Originality:** 4
**Overall Recommendation:** 4
**Confidence:** 4

**Summary:**

This work studies the computational properties surrounding the gradient of place field widths in hippocampus along the dorsal-ventral axis. In particular, the authors: 1) show how this diversity can be generated by changes in connectivity to grid cells in different modules; 2) how this diversity can enable good performance across tasks in the low-data limit. In addition, the authors use their results to inspire a new approach for embedding in Transformers.

**Compliance With Llm Reviewing Policy:**

Affirmed.

**Final Justification:**

The authors have addressed my questions and I now feel more confident that this work is sufficient to be accepted.

**Key Questions For Authors:**

Before I can recommend accepting this paper, I would like the following answered:

1. How do the results shown in Fig. 7 fit with the theory from Eq. 13? Why does $\sigma = 20$ not get the best performance of $D = 21$?

2. How did the authors implement the $\sigma$ values in the Transformer? Did they only use one value?

**Limitations:**

The authors did a sufficient job addressing limitations. I think it could be interesting and beneficial if they considered what happens if dorsal hippocampus does not receive projections from ventral MEC.

**Strengths And Weaknesses:**

## Strengths:

1. The authors study an interesting problem - the gradient of place cell widths - at several different levels. Their results on inspiring new methods for embedding in Transformers (including Language Models) is a great illustration of what neuro-AI and computational neuroscience offers the field of machine learning.

2. The authors provide precise characterization of their model, which allows for a detailed understanding of: 1) how place cells of different widths can be generated; 2) how place cell width impacts performance.

3. The authors make a specific prediction on anatomy of grid-to-place cells.

4. The use of both synthetic and real-world data for the Transformer was good and helped provide intuition + evidence that the method really could work in applied problems

## Weaknesses:

1. The authors find that, in the few-shot learning regime, the optimal place-cell width is proportional to $D$, the size of the task-region. This comes from the fact that the optimal place-cell width is proportional to $L n^{-1}$. I don't think $n$ was ever defined (I assume this is the number of regions?), so this step was not explicitly clear. More importantly though, in Fig. 7 the authors test populations of different fixed place-cell widths on different numbers of regions $D$. If I understood Eq. 13 correctly, I would expect $\sigma = D$ to be optimal in Fig. 7. However, $\sigma = 5$ does much better on $D = 21$ than $\sigma = 20$. This then seems like a contradiction to the authors analytical work.

2. The results from Sec. 3.4 seem to motivate an embedding in the Transformer with multiple $\sigma$ values. But when discussing the results of Experiment 1 (Sec. 5.2), the authors mention specific $\sigma_p$ values. Does this mean they only used one place cell width? This then is also confusing when the authors discuss the ablation where they only use value of $\sigma$ (Table 3).

3. I think overall,  Secs. 4 and 5 could use more explicit detail (maybe a schematic illustrating exactly what was done). In particular, it was not clear to me: 1) what $\sigma$ values were used and if there was a gradient (if so, how exactly was this implemented); 2) why the authors used a bipartite hierarchical structure - I can see from the ablations that this worked well, but what was the motivation for keeping the first several layers with the original embedding?

4. The referencing was a point of weakness. The authors mention "While prior theoretical work has established grid cells’ mathematical role in spatial coding", only one paper is cited. This ignores decades of work on grid cell theory (e.g., work by Ila Fiete, Xuexin Wei, Andreas Herz, Tim Behrens, Surya Ganguli, ...). The authors use a linear transformation of grid-to-place cells, but never mention why they do so (e.g., what experimental evidences supports this transformation? - there is a lot of it, but with the exception of Mallory et al. 2018, this is not referenced). In addition, the authors do not cite the original paper that consider grid-to-place transformation through a linear mapping (Solstad et al., 2006). The authors mention high-precision encoding of position by place cells, but do this with a paper from 1993. A more recent, and more complete account of this can be found in Hazon et al. (2022).


## Minor points:

1. $\kappa$ is defined with $\kappa$ on both the left and right hand sides of the equation.

2. The authors make explicit predictions about anatomical connectivity between MEC and HPC. It is my understanding that long range projections from ventral MEC to dorsal hippocampus is not so likely (perhaps this is incorrect on my part - but see van Strien et al. 2009). It would be interesting to know if/how the model is able to compensate for not getting inputs from all modules. If dorsal hippocampus only gets inputs from the most dorsal half of MEC, would that change the predicted synaptic connectivity?

---

> ### Author Rebuttal · Authors · 2026-03-31
>
> **W1&Q1. Definition of $n$ and Figure 7 Performance**
>
> We thank the reviewer for noting that $n$ was not explicitly defined. In our framework, $n$ is the dominant spatial frequency mode, with region width $D = L/(2n)$. We will state this explicitly in the revision.
> For Figure 7, Eq. 13 implies a proportionality ($\sigma_p^{\mathrm{opt}} \propto D$), not a strict equality $\sigma_p = D$. Empirically, Figure 6 shows $\sigma_p^{\mathrm{opt}} \approx D/2$. Thus, for $D = 21$, the optimum is about $10.5$: $\sigma = 20$ is too large and over-smooths the representation, whereas $\sigma = 5.0$ is much closer to the optimum (and $\sigma = 12.5$ is closer still, and performs best). Therefore, the fact that $\sigma = 5$ outperforms $\sigma = 20$ at $D = 21$ is not a contradiction, but is consistent with the approximate scaling predicted by our analysis.
>
> **W4. Relevant Works on Grid Cell Theory**
>
> We thank the reviewer for pointing out these key works. We will expand the introduction to properly situate our study within this rich literature, distinguishing our focus on downstream grid-to-place transformations from works addressing grid code emergence (e.g., Fiete et al., 2008; Wei et al., 2015). We will cite Solstad et al. (2006) for the linear mapping, noting our distinct use of discrete grid modules (Stensola 2012) over logarithm distribution. Finally, we will update our precision encoding citation to Hazon et al. (2022).
>
> **MP1. $\kappa$**
>
> We thank the reviewer for catching this. $\kappa$ is a self-consistency variable; we will clarify this in the revision.
>
> **MP2. Anatomical Connectivity Constraints**
>
> We thank the reviewer for highlighting the established anatomical topography. In our framework, generating broad place fields requires low-frequency grid inputs. If the dorsal HPC only receives dorsal MEC inputs, it is restricted to higher spatial frequencies, though these still reach scales up to ~1m (Stensola et al., 2012). Consequently, when the environment is smaller than the available grid scales (e.g., ~50cm), these inputs provide sufficient frequency coverage. Under these conditions, the model naturally compensates without missing necessary modules, and the predicted synaptic connectivity decay remains mathematically identical.
>
> **W2&W3&Q2. HIPE implementation and bipartite design**
>
> Thank you for pointing this out. We agree that Sections 4–5 did not describe the HIPE implementation explicitly enough.
>
> HIPE is a general multiscale framework: in principle, different layers and different heads can use different $\sigma$, and standard RoPE is recovered as the special case $\sigma\to0$. Thus, HIPE is not restricted to a single place-field width. In the main experiments of the current paper, however, we adopted a controlled layer-wise schedule.
>
> Concretely, in the bipartite setting, the shallow layers use the $\sigma\to0$ limit (the RoPE special case within HIPE), while the deeper layers use a larger nonzero $\sigma$. We chose this design as a simple controlled instantiation for testing the hypothesis of early short-range positional modeling and later broader long-range integration, and also because it makes it easier to search for an appropriate $\sigma$ scale for different sequence lengths without introducing too many free variables at once.
>
> Importantly, keeping the shallow layers at $\sigma \to 0$ is not merely reusing the baseline, but a deliberate choice within the HIPE framework to preserve fine-grained positional discrimination in early layers. We further validated this motivation with a new local-global attention synergy experiment (60M model, 300M tokens, C4 dataset, context length 1024, G means global attention, L means local attention):
>
> |Experiment|Shallow Layers (0-3)|Deep Layers (4-7)|Perplexity|
> |-|-|-|-|
> |**Baseline**|G+HIPE, $\sigma\to0$ (RoPE)|G+HIPE, $\sigma\to0$ (RoPE)|62.82|
> |**Local-Global**|L+HIPE, $\sigma\to0$ (RoPE)|G+HIPE, $\sigma\to0$ (RoPE)|59.36|
> |**Global-Local**|G+HIPE, $\sigma\to0$ (RoPE)|L+HIPE, $\sigma\to0$ (RoPE)|64.32|
> |**Bipartite HIPE**|G+HIPE, $\sigma\to0$ (RoPE)|G+HIPE, $\sigma=500$|62.71|
> |**Global-Local HIPE**|G+HIPE, $\sigma\to0$ (RoPE)|L+HIPE, $\sigma=500$|64.30|
> |**Local-Global HIPE**|L+HIPE, $\sigma\to0$ (RoPE)|G+HIPE, $\sigma=500$| **59.01**|
>
> These results support the functional interpretation behind our bipartite design: shallow layers benefit most from short-range positional modeling, whereas deeper layers require global access and broader integration.
>
> Finally, we also tested learnable-$\sigma$ (300M model, 1B tokens, C4 dataset), which further improved over fixed-$\sigma$ HIPE (29.04 $\to$ 28.97 at len=512; 31.93 $\to$ 31.69 at len= 2048 for 300M model). This supports our broader point that the bipartite schedule in the current paper is only the simplest controlled instantiation of HIPE, rather than the only possible implementation.
> |**Model**|**PPL@Len=512**|**PPL@Len=2048**|
> |-|-|-|
> |Fixed $\sigma$|29.04|31.93|
> |Learnable $\sigma$| **28.97**| **31.69**|

---

> > ### Author Rebuttal · Reviewer_n6qN · 2026-04-02
> >
> > I thank the authors for their responses. My questions have been satisfactorily answered.

---

### Official Review · Reviewer_FeeH · 2026-03-13

**Soundness:** 3
**Presentation:** 3
**Significance:** 2
**Originality:** 2
**Overall Recommendation:** 4
**Confidence:** 3

**Summary:**

The hippocampus (HPC) presents a gradient of place field sizes along the dorso-ventral axis. Those place fields are thought to be generated by a linear recombination of grid cells in entorhineal cortex (EC), themselves presenting a (modular) gradient of frequencies along the dorso-ventral axis. This study first determines the connectivity between EC and HPC that could give rise to the HPC gradient. It then proposes a normative justification for the coexistence of place fields of different sizes: In a task with varying relevant spatial frequencies and varying sample sizes, it shows that small fields allow for high-fidelity learning for large sample sizes and/or rapidly varying landscapes, whereas large fields enable few-shot learning in slowly evolving landscapes. The finding is backed up by experiments and theory of linear generalization. Finally they propose a bio-inspired place coding transform for transformers with varying receptive field size, showing benefits on various tasks.

**Compliance With Llm Reviewing Policy:**

Affirmed.

**Final Justification:**

The rebuttal adequately addressed my clarity concerns. I thus maintain my positive score, which however also continues to reflect my slight reservations aforementioned in the review regarding Originality and Significance and which cannot be addressed in a short rebuttal.

**Key Questions For Authors:**

In 3.1, The model architecture should be specified in the text, ideally with equations. Is it a linear model, as suggested by fig 3? If yes, how can it do an AND operation on sensory inputs?

Would the tradeoff between generalisation and spatial precision observed also be observed if the readout was non-linear? In the absence of noise source, and with a non-linear readout, I believe a priori that many size place fields should allow perfect localization, and so perfect solving of the task.

In 3.3, y(x) is the target function. Is this target function still the AND/OR task on sensory input? But then how is it possible that the first equation in 3.3 does not use sensory inputs, only a linear readout of place cells?

Σ(x, x′ ) = p(x)⊤ p(x) => x' missing in right-hand side of equation

Experiments of 3.4 are compelling but details are missing. Is it still the AND/OR task? And is it in the few-shot regime? The 5-5 place field size (small place fields) performs almost as well as 5-20 which suggests a not so sparse regime.

eqn 15 define q and k, key and query

RoPE consists in a rotation applied to tokens, and seems hardly analog to the place fields found in HPC. Such place fields seem more directly analog to positional embeddings used in transformers. Moreover, the bioinspiration of HIPE seems weak. Whereas in HPC there is co-existence of many spatial scales, here the only change to RoPE is to smear high frequencies of the positional encoding in specific layers, leading to better results in the specific experiment presented. "This confirms that the model dynamically adapts its spatial inductive bias: " => There is no mechanism of adaptation proposed here, it is just a grid search.

**Limitations:**

yes

**Strengths And Weaknesses:**

Soundness: Claims are well supported by the data. Methods used are appropriate. Theoretical results are sound and correct as far as I can tell.

Presentation: The article is overall clear and well-written, although it left me with some questions regarding the details of the experiments and theory (see Questions).

Significance: The significance of the results appears somewhat weak. Although it is presented as a paradox that a smooth gradient of place fields can be obtained from the linear recombination of a discrete number of frequency fields, it is hardly a novel insight from the perspective of spectral theory. The proposed normative justification for the coexistence of place fields of different scales is interesting, and the connection with theory of linear generalization fitting, although no optimality of HPC arrangement is shown. It is hard for me to judge of the importance of the machine learning finding, but see Originality.

Originality: The results present some interesting insights into the potential connectivity between EC and HPC, as well as on the computational benefits of the gradient of place field sizes. From the machine learning point of view, however, the proposed positional embedding at multiscale resolution hardly seems like a novel idea, and seems only tangentially related to the biological finding, insofar as the architecture proposed adapts RoPE to have only one spatial scale per layer, as opposed to the coexistence of the gradient of scales found in HPC (see also related Question).

---

> ### Author Rebuttal · Authors · 2026-03-31
>
> **Model architecture, AND/OR implementation, and generalization error (Sec 3.1 & 3.3)**
>
> We thank the reviewer for highlighting these points. The readout is indeed linear: $\hat{y}(x) = \mathbf{w}_s \mathbf{s}(x) + \mathbf{w}_p \mathbf{p}(x)$, where $\mathbf{s} \in \{0,1\}^2$ are binary sensory inputs. Both AND and OR are linearly separable over $\{0,1\}^2$ (e.g., AND equates to $s_1 + s_2 - 1.5 > 0$). The spatial position $\mathbf{p}(x)$ acts as a gate, determining which linear operation applies.
>
> Regarding Section 3.3, $y(x)$ remains the AND/OR task target. Because $\mathbf{s}(x)$ and $\mathbf{p}(x)$ are orthogonal by construction, the generalization error decomposes explicitly as:
> $$
> E_g=\langle (\mathbf{w}_s \mathbf{s}(x)+\mathbf{w}_p \mathbf{p}(x) - y(x))^2 \rangle=\langle(\mathbf{w}_p\mathbf{p}(x)-y(x))^2\rangle - \langle\mathbf{w}_s\mathbf{s}(x) (\mathbf{w}_s\mathbf{s}(x)-2y(x))\rangle
> $$
> The sensory weights $\mathbf{w}_s$ can perfectly fit the task's sensory component, so the remaining generalization error is driven entirely by the place code term $\langle(\mathbf{w}_p\mathbf{p}(x)-y(x))^2\rangle$. This is why Section 3.3 exclusively analyzes this term. We will add this decomposition to the revision and correct the covariance typo to $\Sigma(x,x') = \mathbf{p}(x)^\top\mathbf{p}(x')$.
>
> **Precision/generalization trade-off with a non-linear readout**
>
> We appreciate this constructive suggestion and evaluated a non-linear readout, $\hat{y} = \tanh\left(\mathbf{W} \cdot \mathrm{concat}(\mathbf{s}, \mathbf{p}(x))\right)$, trained over 300 epochs with learning rate = 0.01. As shown below, $\sigma_p = 15$ consistently yields the highest accuracy. This confirms the trade-off is a fundamental property of the representational spectral structure. While expressive non-linear readouts achieve perfect accuracy with infinite data, the sample efficiency advantage of optimally-sized place fields remains robust in the few-shot regime. We will include these results in the revision.
>
> |Sample size $S$ \ Place field size $\sigma_p$|5.0|11.0|15.0|21.0|25.0|
> |:-|:-|:-|:-|:-|:-|
> |**16**|0.664|0.729|**0.749**|0.684|0.689|
> |**32**|0.756|0.833|**0.865**|0.767|0.741|
> |**64**|0.848|0.915|**0.940**|0.826|0.769|
> |**128**|0.911|0.956|**0.970**|0.877|0.787|
>
> **Experimental details and scale comparisons in Section 3.4**
>
> Yes, Section 3.4 uses the same AND/OR task in the few-shot regime ($S = 16$, as labeled in Figure 7).
>
> Regarding the performance: the $(5,5)$ and $(5,20)$ models only perform similarly on small region widths, where narrow fields happen to be optimal. However, as the region width grows, the $(5,5)$ model's accuracy degrades substantially. Consequently, the multi-scale $(5,20)$ model achieves a significantly higher average accuracy across all environments ($p < 10^{-4}$). This validates that a multi-scale architecture is essential because no single field size is optimal across all spatial scales.
>
> **Eq. 15 notation (q, k as query and key vectors)**
>
> We thank the reviewer for catching this. Eq. 15 introduces q and k without defining them. We will revise Section 4.1 to state explicitly that they are the query and key vectors in attention. This is a notation fix only and does not affect the derivation.
>
> **Biological analogy between RoPE, HIPE, grid codes, and place fields**
>
> We do not claim that token positions are literally the same as physical space in the hippocampus, or that Transformer positional encoding is a neural model of place cells. The similarity is at the level of computation: both systems need a positional code that supports multi-scale retrieval, balancing local precision with broad integration.
>
> In the hippocampus, this is achieved by frequency-dependent integration over grid-like bases, producing place fields of different widths. HIPE transfers the same multi-scale weighting principle to Transformer attention. Within this framework, standard RoPE corresponds to the $\sigma\to0$ limit of HIPE. Thus, our biological inspiration is not a claim of literal neural equivalence, but of a shared positional-coding problem and a transferable inductive bias.
>
> We also agree that the current bipartite design is only a controlled approximation of hippocampal multiscale coexistence, not a literal reconstruction. It was designed so that shallow layers focus on short-range positional information, while deeper layers support long-range integration. This bipartite setting gives a simple and effective way to implement that functional hierarchy, while avoiding a full search over all layers and heads.
>
> **"Dynamically adapts spatial inductive bias"**
>
> We agree this wording is too strong. The current paper does not include an explicit adaptation mechanism. Our results only show that the best preset spatial scale increases with sequence length, consistent with the theory that broader fields are favored in larger environments. We will revise Section 5.3 accordingly and replace “dynamically adapts” with a more precise statement.

---

> > ### Author Rebuttal · Reviewer_FeeH · 2026-04-03
> >
> > I thank the authors for their clarifications which adequately address my clarity concerns. I maintain my positive score, which also continues to reflect my reservations aforementioned.

---

### Official Review · Reviewer_hcFE · 2026-03-14

**Soundness:** 2
**Presentation:** 2
**Significance:** 2
**Originality:** 3
**Overall Recommendation:** 3
**Confidence:** 3

**Summary:**

The aper analytically explore a mechanism to explain the continuously varying sizes of place fields in the hippocampus. With this they make predictions about the functional role of the place fields gradient (explaing the tradeoff between precision and generalization).  The authors also provide predictions for anatomical projection patterns, as well as on optimal estimation for place field sizes regarding the environment.

Based on this analytical framework authors propose a biologically inspired positional embedding method for transformers (HIPE), which is an extensional of positional embedding method with selective benchmarking comparing conventional RoPE methods. The authors make conclusion regarding the contribution of the this method for sample efficiency learning.

**Compliance With Llm Reviewing Policy:**

Affirmed.

**Key Questions For Authors:**

1) in the design of HIPE, are the authors assuming the grid like attention heads are completely orthogonal to each other ? If so, this assumption is not very valid in biological circuit so they might need to show the consequence of overlapping grid frequencies in the HIPE design.

2) How did the author chose the exponential parameters of the projection connectivity regarding frequency of the grid untis in the HIPE design?

3) How extendable authors find HIPE algorithm and to larger models ?

**Limitations:**

1) Although the core claim of the paper is about a bio-inspired adjustment to attention heads for sample efficiency, the lack of comparison to the state of the art extension from positional architectures ( Like Hyena and YaRN) would leave the specific superiority of the method ambiguous.

2) The novelty of the paper to heavily rely on biological assumption, yet the proposed framework is not tested directly on real or even synthetic neural data resembling place and grid cells.


3) the codes for the implications are lacking. (from what I see from submission file)

**Strengths And Weaknesses:**

Strengths:
The intuition behind the design of the hipe method is biologically elegant and authors have tried to test the proposed method on some benchmark.

Weekness:
The proposed theoretical section has been very finely designed to explain the biological properties of place fields observed in hippocampus, however it is not clear which of those designs choices are necessary or sufficient for the success of HIPE in benchmark tasks.

The implementation of the HIPE method on transformers seems intuitive however without the comparison with recent adjustments of Rope family that are more successful in recent years (like YaRN: https://arxiv.org/abs/2309.00071)  it's hard to make a conclusion about added value of HIPE method beyond biological inspiration, eventhough it is better than early papers adjustment of positional embedding.

---

> ### Author Rebuttal · Authors · 2026-03-31
>
> **W1**
>
> Thank you for this important point. We agree that the current draft does not clearly separate which parts of the biological framework are the main algorithmic ingredients for HIPE’s benchmark performance.
>
> Our current evidence suggests that the empirical gains are mainly driven by two factors: (1) frequency-dependent spectral scaling, which turns positional spectrum into a multiscale positional bias, and (2) its sharp-to-broad deployment across layers, which preserves fine positional sensitivity in shallow layers while enabling broader integration in deeper layers.
>
> So our intended claim is not that every biological design choice is itself necessary or sufficient for benchmark gains. Rather, the theoretical section provides a principled source of multiscale positional bias, while the benchmark results suggest that the key ingredients for HIPE’s empirical performance are the induced spectral scaling and its hierarchical deployment across layers. We will revise the paper to make this distinction clearer.
>
> **Q1&Q2**
>
> We thank the reviewer for these points. Rather than assuming discrete orthogonal grid modules, our framework models the positional encoding over a continuous frequency spectrum. In infinite space, Fourier basis functions are orthogonal by construction, so no additional orthogonality assumption is required. Even in the biologically realistic case where grid modules have finite tuning width and partially overlap, their collective effect is well approximated by a smooth spectral density $\rho(\omega)$, handled naturally within the continuous framework.
>
> Regarding the choice of frequency parameters, we retain RoPE's standard exponential schedule $\omega_j = 10000^{-2(j-1)/d}$ for direct comparability with existing baselines. This schedule induces a non-uniform sampling density $\rho(\omega) \propto 1/\omega$, which is corrected by the $\sqrt{\omega}$ Jacobian term in the HIPE scaling factor:
> $$
> S(\omega) = \lambda\cdot\sqrt{\omega\cdot
>     \exp\left(-\frac{\sigma^2\omega^2}{2}\right)},
> $$
> where the $\sqrt{\omega}$ term compensates for the non-uniform frequency density, ensuring the discrete dot-product sum correctly approximates the continuous Gaussian place field integral, and
> $\exp(-\sigma^2\omega^2/2)$ implements the frequency-dependent decay derived from the grid-to-place projection.
>
> **W2&Q3&L1**
>
> Thank you for this important suggestion. We agree that the original draft did not provide enough evidence to establish the added value of HIPE beyond biological motivation, especially without comparison to newer positional baselines or validation at a larger scale.
>
> In our framework, standard RoPE corresponds to the sharp limit $\sigma \to 0$. The paper states this explicitly in both the section 4.1 and appendix C.
> This is important for interpreting the new results: we do not expect broader-scale HIPE to always beat RoPE, because RoPE is already the finest-scale limit of the same family.
>
> Larger-scale pretraining: HIPE stays competitive, but the gap to RoPE becomes small. We added experiments with a 300M model trained on 1B tokens on C4.
>
> |Model|PPL@512|PPL@2048|
> |-|-|-|
> |RoPE|**28.93 ± 0.031**|**31.68 ± 0.039**|
> |HIPE (learnable $\sigma$)|28.97 ± 0.032|31.69 ± 0.024|
> |HIPE (fixed $\sigma$)|29.04 ± 0.043|31.93 ± 0.030|
> |XPos|29.97 ± 0.055|32.75 ± 0.063|
>
> These results show that HIPE remains competitive with RoPE and also performs better than XPos, a added RoPE-style baseline with frequency-dependent decay, in our setup. At the same time, the gap between HIPE and RoPE becomes very small at this larger scale. This is consistent with Figure 4: when data become abundant, the sharper regime can become more favorable.
>
> Using the same 300M pretrained models, we ran LoRA fine-tuning on SST-2 ACC with limited labeled data:
>
> |Samples|RoPE-512|HIPE-512|
> |-|-|-|
> |100|67.3 ± 5.0|**68.5 ± 2.0**|
> |200|68.8 ± 6.4|**70.1 ± 5.5**|
> |500|75.7 ± 0.1|**76.4 ± 1.6**|
> |1k|76.3 ± 0.6|**77.4 ± 0.3**|
> |2k|77.9 ± 0.3|**78.2 ± 1.5**|
> |full (67k)|**82.0 ± 0.40**|81.8 ± 0.65|
>
> This again matches the paper’s main point from Figure 4: broader-scale settings help more when supervision is limited, while the sharp RoPE limit can again become favorable when enough labeled data are available.
>
> We also compared with YaRN in a length-extrapolation setting (train at 512; test at 1024/2048/4096):
>
> |Sequence Length|YaRN|HIPE|
> |-|-|-|
> |Extrap (1024)|**30.93**|31.89|
> |Extrap (2048)|69.10|**62.01**|
> |Extrap (4096)|128.15|**116.81**|
>
> In our setting, YaRN is better at length=1024, while HIPE degrades more slowly at more extreme extrapolation lengths. We therefore view them as having different inductive-bias profiles, rather than one uniformly dominating the other.
>
> **L2**
>
> Thank you for this important point. We agree that the current paper does not directly test the framework on neural data. Our focus here is theory-to-algorithm translation, and we will clarify this more explicitly.
>
> **L3**
>
> https://anonymous.4open.science/r/PE-F0E0/

---

> > ### Author Rebuttal · Reviewer_hcFE · 2026-04-04
> >
> > Thanks for the explanation and sharing the code. In light of author’s clarification I increase my confidence to 4 and remain my overall borderline assessment.

---

> > > ### Author Response · Authors · 2026-04-07
> > >
> > > Dear Area Chair and Reviewer hcFE,
> > >
> > > As the rebuttal phase concludes, we would like to provide a final clarification on the fundamental positioning of our work to support the AC’s final assessment.
> > >
> > > We respect the reviewer hcFE’s perspective from a large-scale model. However, as categorized under "neuroscience and cognitive science", our goal is to elucidate the mechanism of the projection from grid cells to place cells, explore its functional benefits, and build a theoretical bridge between hippocampal modeling and positional embedding in Transformer architecture.
> > >
> > > 1. **Theoretical Migration vs. Engineering Update:** Our work proposes a mathematical framework moving from anatomical connectivity (synaptic weights) to multiscale neural representations, and ultimately to behavioral sample efficiency, establishing a **unified positional coding strategy for bio and artificial intelligence**. In both the hippocampus and HIPE, a single parameter (the place field size $\sigma$, representing the spatial attention scale) incorporates biological inductive biases to solve the challenge of **sample-efficient learning**—a domain where biological brains still outperform current AI.
> > > 2. **Validation of Theory at Scale:** Following the reviewer's suggestion, our new experiments at the 300M-parameter/1B-token scale have confirmed our theoretical prediction: larger $\sigma$ values (broader fields) significantly aid generalization in low-data regimes, while standard RoPE (as HIPE $\sigma \rightarrow 0$) excels when data is abundant. This "spectrum of trade-offs" is precisely what the hippocampal gradient manages.
> > > 3. **Biological Mapping:** To resolve concerns about the biological inspiration of HIPE, the final version will mathematically formalize the attention logit $(q_n​k_m^T​∝e^{−(m−n)^2/2\sigma_{p}^2}​)$ as a localized place field centered at token position $n$. Integrating the semantic value vector ($v_m$​) directly mirrors the conjunctive coding of place cells, which bind spatial locations to sensory inputs. This grounds HIPE as a mathematically shared computational solution across biological and artificial intelligence.
> > >
> > > In summary, we believe HIPE offers a bio-inspired perspective on the interpretability and efficiency of positional embeddings. We hope the Area Chair recognizes the value of this neuroscience-to-AI migration, which Reviewer 5eFt aptly highlighted as a "quintessential form of neural computation" that is "rare but appreciated" in the field.
> > >
> > > Thank you both for your rigorous and constructive engagement with our work.
> > >
> > > Best regards,
> > >
> > > The Authors

---

### Official Review · Reviewer_5eFt · 2026-03-23

**Soundness:** 3
**Presentation:** 3
**Significance:** 3
**Originality:** 3
**Overall Recommendation:** 4
**Confidence:** 3

**Summary:**

The article synergizes insights from neuroscience to develop a biologically-inspired approach for positional encoding in transformer neural networks. First, the paper walks the reader through details of place cell responses in hippocampus (HC), and the organization of grid cell responses in medial entorhinal cortex (mEC), a known primary input area to the HC. While grid cells have a modular organization – with a discrete set of frequencies represented – place cells in the CA1 subregion of HC exhibit a continuous topographic gradient of place field size (i.e., the size of the spatial area they are sensitive to), increasing in size preference from dorsal to ventral CA1. The authors aim to answer the question of how the hippocampus transforms inputs from the discrete set of frequencies represented in mEC to the smooth gradient in HC. They develop a mathematical model of the emergence of Gaussian-like hippocampal place fields from a discrete set of periodic sin/cosine functions representing grid cells, using a 1D simplification. Via several simplifications (an assumption that place fields are sufficiently small relative to the environment, assumption of linearity, assumption that place fields arise solely from grid cell inputs) the authors arrive at a closed form solution to recover the weights from grid cells onto place cells. This function exhibits gaussian decay of weights onto grid cells as a function of spatial frequency, with larger place fields decaying more quickly. Based on this organization, the authors transition to asking functional questions related to behavior: what significance does the gradation of place field size have for behavior? The authors introduce a context-dependent computation task, where an agent must perform an “AND” or “OR” operation depending on its spatial position, testing data-limited and data-rich regimes. They find interesting relationships between test accuracy, sample size, and place field size. Briefly, small place fields support fine-grained behavior, while large place fields support generalization in lower data regimes. In general, agents with a distribution of place field sizes perform better than agents with a single characteristic place field size. Based on these insights, the authors hypothesize that variable spatial tuning can benefit transformers via the positional encoding. The authors describe standard RoPE as the “dorsal limit” – where spatial sensitivity is extremely precise. They then introduce “hippocampous inspired positional embedding” (HIPE), which aims to inject a particular larger degree of spatial tuning into RoPE, in an efficient manner that introduces minimal computational overhead and plays nice with modern optimized transformer libraries (e.g. flash attention). Using small-scale simulations of transformers (20M-60M models, .1B tokens), the authors demonstrate favorable performance of HIPE relative to RoPE, ALiBi, and NoPE. Notably, HIPE shows strong performance in the low-data regime, likely owing to its larger positional tuning in later layers that aid generalization, and is favorable relative to ALiBi at long sequence lengths due to its superior amenability to efficient implementation.

**Compliance With Llm Reviewing Policy:**

Affirmed.

**Key Questions For Authors:**

Questions
-	What aspect, if any, of the mathematical model introduced in section 2 is novel? From my understanding, it has been well known that place cells can be explained via a linear combination of grid cell responses for nearly 20 years. The paper presents this as a seemingly novel contribution, but it is unclear how if at all this is true.
-	How significant is the simplification of the integration from 0 to L to -inf to +inf for the general motivation and conclusions of the paper? In my opinion, the simplification is not particularly justified because there are likely to be place cells near the boundary whose responses would require integration outside of the boundary locations. Additionally border cells are a significant input to place cells and are not discussed.
-	How does the bipartite HIPE model explored here compare to models that use local attention in early layers and global attention in later layers? In general, it would be helpful if the authors could discuss HIPE alongside local attention methods, and discuss whether the might be some synergy between the two.
-	Would it be possible to introduce differential spatial tuning within a HIPE layer while maintaining the computational efficiency of the current implementation? What about allowing the tuning to be learned?
-	Would it be possible to run larger scale simulations? The paper could provide significantly more signal to the community if the simulations were run at a larger scale (e.g. .5-1B+ models, >10B tokens).  I am also sympathetic to compute limitations, so the authors can mention if they are strongly compute limited. But it would be good to know that there is not significant performance degradation at more realistic token budgets. It would be interesting to know whether the larger spatial tuning perhaps provides diminishing returns at larger data budgets, but still aids fine-tuning and few shot generalization from larger scale pretrained models.

**Limitations:**

Yes

**Strengths And Weaknesses:**

Soundness: The paper is technically sound. Some of the derivations do rely on simplifying assumptions that are likely not valid (grid cells are the only inputs to place cells, c.f. border cells, linearity of integration, place field size relative to environment), however, I do not think this is critical to the general ideas motivated by the mathematical model, nor do they have any baring on HIPE. Overall, the paper combines mathematics, simulations, neuroscientific insight, and machine learning experimentation in a way that is relatively rare but appreciated within the field of computational neuroscience.

Presentation:
- The paper is overall easy to read, however I feel that the derivations are sometimes a bit cryptic and would do with a bit of revision for clarity and tighter linking to fuller descriptions in the appendix. The authors should clearly check that variables are introduced before appearing in equations, and that the critical steps in derivations are explained.
- The authors describe an “exponential” relationship between weights and spatial frequency; “gaussian” would be more precise, since the exponential is a function of the square of spatial frequency.
- I commend the authors on a truly integrative piece of science – in my opinion, this type of work is the quintessential form of “neural computation”, blending neuroscientific insight, mathematical modeling, and translation into modern performant artificial intelligence

Significance: Many will find this paper insightful and interesting, providing cross-discipline insight that can lead to even more performant neuro-inspired architectures in the future. The results are only suggestive currently, but I think they are still significantly interesting in the context of the broader integrative insights of the paper. Ideally, the authors would strengthen their empirical results to make the paper even more broadly relevant.

Originality: The paper appears to be quite original. It is unclear to me how original any of the math or insights are in the introductory sections. However, I found section 3 to be very interesting and serve as strong motivation to HIPE, which seems to be an original variant of RoPE that is certainly worth exploring and shows some promise.

General weaknesses: the experimental results of HIPE are somewhat underwhelming. First, the paper motivates a graded organization of place fields in HC and spends significant time discussing this smooth organization before introducing HIPE. However, the majority of experiments with HIPE explore a discrete variant where each layer has a single spatial tuning, with early layers having a more fine-grained spatial tuning, and later layers having a more coarse-grained tuning. To the authors credit, they do explore other variants, but it appears the only justification for selecting the main variant is that it works best. For some reason, the other variants with more continuous gradient are not referred to as HIPE (see Table 3).

---

> ### Author Rebuttal · Authors · 2026-03-31
>
> **W**
>
> We agree our terminology was imprecise. HIPE is a unified multiscale positional encoding framework, with RoPE as the $\sigma\to0$ limit; thus, the continuous-gradient, uniform, and bipartite schedules in Table 3 are all HIPE variants. We used the bipartite setting not because the others fall outside HIPE, but because in our current 8-layer, low-data setting it most cleanly instantiates the intended hierarchy: sharp positional sensitivity in shallow layers and broader integration in deeper layers. We will revise the paper to make this explicit.
>
> **Q1**
>
> We agree that linear grid-to-place mapping is well established. Our novelty is in using it to: (1) derive a continuous place-field gradient from discretely scaled grid modules, (2) generate testable predictions for dorsal–ventral heterogeneity in MEC-to-hippocampal projections, and (3) connect these principles to RoPE spectral scaling in Transformers.
>
> **Q2**
>
> Thank you for this observation. The relative error for a place cell at distance ddd from the boundary decays exponentially:
> $$
>     \frac{\left|W_{ij}^{\text{exact}} - W_{ij}^{\text{approx}}\right|}
>     {|W_{ij}^{\text{approx}}|}
>     \leq
>     \frac{\int_{d}^{\infty} e^{-\frac{u^2}{2\sigma_p^2}}du}
>     {\int_{0}^{\infty} e^{-\frac{u^2}{2\sigma_p^2}}du}
>     \approx \frac{\sigma_p}{d}\, e^{-\frac{d^2}{2\sigma_p^2}},
> $$
> The fraction of affected cells is $\sim \sigma_p/L$. For our parameters ($\sigma_p = 5-20$, $L = 100$), this impact is localized to $\leq 20\%$ of the population. This approximation is crucial for achieving the closed-form expression ($|W_i(k)| \propto \exp[-\frac{\sigma_p^2(2\pi k/L)^2}{2}]$) necessary for our theoretical framework.
> Regarding border cells, their inclusion is currently analytically intractable as they lack a quantitative firing model equivalent to our trigonometric grid formulation. We will explicitly state this limitation and the regime of validity in our revision.
>
> **Q3**
>
> We compared HIPE with local/global attention under the same budget (60M, 300M tokens, context length 1024). Local-Global outperforms the baseline (59.36 vs. 62.82), Global-Local performs worse (64.32), and Local-Global HIPE performs best overall (59.01). This shows that HIPE and local/global attention are complementary: local/global attention changes connectivity, while HIPE reshapes the positional spectrum. It also supports the same hierarchy used in our bipartite design: shallow layers benefit from local/fine-grained modeling, while deeper layers benefit from global access and broader integration. You can see the table in Reviewer n6qN's W2&W3&Q2.
>
> **Q4**
>
> Yes. Because HIPE only rescales query/key subspaces, different $\sigma$ values can be assigned to different heads or frequency groups while preserving standard dot-product attention. Thus, the bipartite schedule is only one simple instantiation. We also tested learnable $\sigma$ (300M, 1B tokens, C4), which improves over fixed $\sigma$ (28.97 vs. 29.04 at length 512; 31.69 vs. 31.93 at length 2048), suggesting that adaptive spatial tuning is a promising extension.
> |**Model**|**PPL@Len=512**|**PPL@Len=2048**|
> |-|-|-|
> |Fixed $\sigma$|29.04|31.93|
> |Learnable $\sigma$| **28.97**| **31.69**|
>
> **Q5**
>
> We extended the study to a larger setting (300M model, 1B tokens on C4). Although we cannot yet reach the 0.5–1B+ / 10B+ scale due to compute constraints, this setting is substantially larger than our original experiments and is sufficient to test whether HIPE changes at a more realistic budget.
>
> We first tested sample-limited downstream adaptation using LoRA fine-tuning on SST-2, the results are ACC:
>
> |Samples|RoPE-512|HIPE-512|
> |-|-|-|
> |100|67.3 ± 5.0|**68.5 ± 2.0**|
> |200|68.8 ± 6.4|**70.1 ± 5.5**|
> |500|75.7 ± 0.1|**76.4 ± 1.6**|
> |1k|76.3 ± 0.6|**77.4 ± 0.3**|
> |2k|77.9 ± 0.3|**78.2 ± 1.5**|
> |full (67k)|**82.0 ± 0.40**|81.8 ± 0.65|
>
> HIPE improves over RoPE in most low-resource settings, while the sharper RoPE regime can again become favorable when supervision is abundant. This again matches the paper’s point from Figure 4
>
> At the same larger scale, pretraining PPL remains very close:
>
> |Model|PPL@512|PPL@2048|
> |-|-|-|
> |RoPE|**28.93 ± 0.031**|**31.68 ± 0.039**|
> |HIPE (learnable $\sigma$)|28.97 ± 0.032|31.69 ± 0.024|
> |HIPE (fixed $\sigma$)|29.04 ± 0.043|31.93 ± 0.030|
> |XPos|29.97 ± 0.055|32.75 ± 0.063|
>
> So at this scale, HIPE remains essentially on par with RoPE in pretraining perplexity, while its direct pretraining advantage becomes smaller. This is consistent with our theory: broader fields help generalization in low-data regimes, whereas narrower fields yield better precision in data-rich regimes. Our updated interpretation is that HIPE primarily acts as a sample-efficiency-oriented multiscale inductive bias.

---

> > ### Author Rebuttal · Reviewer_5eFt · 2026-04-04
> >
> > I am pleased to see the authors have tested out some of my suggestions and found promising results. I remain enthusiastic about the paper despite its somewhat underwhelming empirical advantages, which are now clearly held in the low-data regime. This is not a weakness per se, as I consider this to be a paper of primarily theoretical interest to computational neuroscience and, to some extent, machine learning. As I stated before, I think many will find it interesting. I believe it would be easy for others to build on this work to test variants of it in larger compute settings, as well, and perhaps with some changes it could become more practical, especially in post-training scenarios where there may be less data in a particular subject.
> >
> > I recommend that the authors include the new results mentioned here, and share their code so that others can build upon their interesting findings.

---

### Decision · Program_Chairs · 2026-04-30

**Decision:**

Accept (regular)

**Comment:**

Place cells in the hippocampus are experimentally known to exhibit a gradient of place fields. Authors provide results about how such place fields can arise from entorhinal grid cell inputs and the inductive bias set by the multiscale place field representations. Authors further propose a place-cell inspired positional encoding to "induce multi-scale representations".

Reviewers agreed that the paper was mostly well-written and interesting. A particular strength was the link it makes between neuroscience and practical aspects of training transformers through developing a new kind of positional encoding. Some questions were raised about experimental improvement due to HIPE being modest and choice of benchmarks. During the rebuttal, authors responded to these questions and reviewer confidence and scores were adjusted. Overall, the final assessment was borderline, with a slight accept tilt.